# How Rainfall Events Modify Trace Gas Mixing Ratios in Central Amazonia

Luiz A. T. Machado[1,2], Jürgen Kesselmeier[1], Santiago Botía[3], Hella van Asperen[3], Meinrat O. Andreae[1,19,20], Alessandro C. de Araújo[4], Paulo Artaxo[2], Achim Edtbauer[1], Rosaria R. Ferreira[5], Marco A. Franco[21], Hartwig Harder[1], Sam P. Jones[3], Cléo Q. Dias-Júnior[6], Guido G. Haytzmann[2], Carlos A. Quesada[5], Shujiro Komiya[3], Jost Lavric[3,15], Jos Lelieveld[1,7], Ingeborg Levin[8], Anke Nölscher[1,16], Eva Pfannerstill[1,17], Mira M. Pöhlker[1,9,10], Ulrich Pöschl[1], Akima Ringsdorf[1], Luciana Rizzo[2], Ana M. Yáñez-Serrano[11,12,13], Susan Trumbore[3], Wanda I. D. Valenti[5], Jordi Vila-Guerau de Arellano[14], David Walter[1,3], Jonathan Williams[1], Stefan Wolff[1,18], and Christopher Pöhlker[1]

[1] Max Planck Institute for Chemistry, 55128 Mainz, Germany
[2] Instituto de Física, Universidade de São Paulo, São Paulo, Brazil
[3] Max Planck Institute for Biogeochemistry, Jena, Germany
[4] Embrapa Amazônia Oriental, Belém, PA, Brazil
[5] Instituto de Pesquisas da Amazonia, INPA, Manaus, Brazil
[6] Department of Physics, Federal Institute of Pará, Belém, PA, Brazil
[7] The Cyprus Institute, Climate and Atmosphere Research Center, Nicosia, 1645, Cyprus
[8] Institut für Umweltphysik, Heidelberg University, Heidelberg, Germany
[9] Faculty of Physics and Earth Sciences, Leipzig Institute for Meteorology, University of Leipzig, Leipzig, Germany
[10] Experimental Aerosol and Cloud Microphysics Department, Leibniz Institute for Tropospheric Research, Leipzig, Germany
[11] Instituto de Diagnóstico Ambiental y Estudios del Agua (IDAEA), Barcelona, Spain
[12] Centro de Investigación Ecológica y Aplicaciones Forestales (CREAF), Catalonia, Spain
[13] Global Ecology Unit (CREAF-CSIC-UA), Catalonia, Spain
[14] Meteorology and Air Quality Section, Wageningen University, The Netherlands
[15] Acoem GmbH, Hallbergmoos, Germany
[16] University of Bayreuth, Bayreuth, Germany
[17] Institute of Energy and Climate Research, Jülich, Germany
[18] German Weather Service, 63067 Offenbach, Germany
[19] Department of Geology and Geophysics, King Saud University, Riyadh, Saudi Arabia
[20] Scripps Institution of Oceanography, University of California San Diego, La Jolla, USA
[21] Institute of Astronomy, Geophysics and Atmospheric Sciences, University of Sao Paulo, Brazil

**Correspondence:** Luiz A. T. Machado (l.machado@mpic.de))

**Abstract.** This study investigates the rain-initiated mixing and variability in the mixing ratio of selected trace gases in the atmosphere over the central Amazon rain forest. It builds on comprehensive data from the Amazon Tall Tower Observatory (ATTO), spanning from 2013 to 2020 and comprising the greenhouse gases (GHG) carbon dioxide ($CO_2$) and methane ($CH_4$), the reactive trace gases carbon monoxide (CO), ozone ($O_3$), nitric oxide (NO), and nitrogen dioxide $NO_2$) as well as selected volatile organic compounds (VOC). Based on more than 1000 analyzed rainfall events, the study resolves the trace gas mixing ratio patterns before, during, and after the rain events, along with vertical mixing ratio gradients across the forest canopy. The assessment of the rainfall events was conducted independently for daytime and nighttime periods, which allows us to

elucidate the influence of solar radiation. The mixing ratios of $CO_2$, CO, and $CH_4$ clearly declined during rainfall, which can be attributed to the downdraft-related entrainment of pristine air from higher altitudes into the boundary layer, a reduction of the photosynthetic activity under increased cloud cover, as well as changes in the surface fluxes. Notably, CO showed a faster reduction than $CO_2$, and the vertical gradient of $CO_2$ and CO is steeper than for $CH_4$. Conversely, the $O_3$ mixing ratio increased across all measurement heights in the course of the rain-related downdrafts. Following the $O_3$ enhancement by up to a factor of two, NO, $NO_2$, and isoprene mixing ratios decreased, whereas monoterpene increased. The temporal and vertical variability of the trace gases is intricately linked to the diverse sink and source processes, surface fluxes, and free troposphere transport. Within the canopy, several interactions unfold among soil, atmosphere, and plants, shaping the overall dynamics. Also, the mixing ratio of biogenic VOC (BVOC) clearly varied with rainfall, driven by factors such as light, temperature, physical transport, and soil processes. Our results disentangle the patterns in trace gas mixing ratio in the course of the sudden and vigorous atmospheric mixing during rainfall events. By selectively uncovering processes that are not clearly detectable under undisturbed conditions, our results contribute to a better understanding of the trace gas life cycle and its interplay with meteorology, cloud dynamics, and rainfall in the Amazon.

# 1   Introduction

A precise understanding of the interaction between the atmosphere and the forest is crucial to accurately simulate and predict the effects of climate and land use change in Amazonia. Atmospheric multiphase processes and chemical reactions are mainly controlled by the mixing ratio of reactive gases and governed by rainfall, thermodynamics, and solar radiation (Ravishankara, 1997). Rainfall events can significantly impact these processes by causing sudden and quite strong changes in trace gas mixing ratios over short periods of time (Bertrand et al., 2008). Therefore, it is essential to understand how weather conditions affect the interaction between the atmosphere and the biosphere to develop comprehensive parameterizations for climate models. The Amazon Tall Tower Observatory (ATTO) is a site equipped with state-of-the-art instruments, laboratories and towers. The key element is a tall tower (325 metres) equipped with a wide range of micrometeorological and trace gas sensors. The so-called Instant and Triangle Towers were erected in 2012, each with a height of 80 meters; see Andreae et al. (2015) for a detailed description of the ATTO instrumentation. These towers have been monitoring gas mixing ratios since then, serving as the primary data source for this study. The data collected at ATTO have been instrumental in advancing our knowledge of the atmosphere-biosphere interaction. Numerous studies have been published using ATTO data, contributing to our understanding of processes such as particle formation. (e.g., Wang et al., 2016; Machado et al., 2021; Franco et al., 2022; Pöhlker et al., 2016; Moran-Zuloaga et al., 2018) and long-range transport (e.g., Holanda et al., 2020, 2023), as well as biogenic gases and particles (e.g., Kesselmeier and Staudt, 1999). The journal Atmospheric Chemistry and Physics features a specialized volume devoted to ATTO research (Amazon Tall Tower Observatory (ATTO) Special Issue), comprehensively analyzing the intricate physical, chemical, and biological interactions inherent in the Amazon rain forest.

The factors influencing the temporal variability of greenhouse gas (GHG) mixing ratio have been studied widely. The global trend of the atmospheric growth rate of carbon dioxide ($CO_2$) (Lan et al., 2023) and methane ($CH_4$) (Thoning et al., 2022) has been documented extensively by global measurement networks, revealing accumulation of these trace gases in the atmosphere. The temporal patterns at a particular measurement site are embedded in this global trend but follow seasonal and diurnal patterns specific to their latitude and local drivers. On a tall tower in Siberia, the Zotino Tall Tower Observatory (ZOTTO) site, fluxes and mixing ratios of carbon monoxide (CO), $CO_2$ and $CH_4$ have been monitored for more than a decade, as reported by Panov et al. (2022); Kozlova et al. (2008); Winderlich et al. (2014), showing seasonal and diurnal variations, driven by the growing season and type of vegetation, either Taiga forests, bogs and old river meanders. At the ATTO site, the seasonal pattern of nighttime $CH_4$ peaks at the 79-m level was described by Botía et al. (2020), highlighting the atmospheric conditions for these events and the potential sources from which these $CH_4$ enhancements could originate. Using a Lagrangian model to obtain the background mixing ratio of $CO_2$ at ATTO (79-m), Botía et al. (2022) derived the regional signal (observations - background) of $CO_2$ and found that the amplitude of the seasonal cycle was about 4 ppm. In that study, they also show how the atmospheric record captured the $CO_2$ anomalies caused by the 2015/2016 El Niño-induced drought. At the diurnal cycle scale, the changes in atmospheric trace gases result from the interaction between local factors and the dynamics and subsequent growth of the planetary boundary layer (PBL). For gases such as $CH_4$, ozone ($O_3$), and nitrogen dioxide ($NO_2$), Mikkelä et al. (1995) found

a relationship between soil temperature, solar radiation, and diurnal emission variability. $CH_4$ exhibits inter-annual variability
and a trend of increasing mixing ratio, the background of which is still not understood (Rigby et al., 2017; Schaefer et al.,
2016; Nisbet et al., 2019). Williams et al. (2001) conducted some of the earliest studies on time-space variability of trace gases
in the Amazon forest and observed large spatial and temporal variability of gas mixing ratio, resulting in strong gradients of
CO and $CO_2$ mixing ratios. Given these studies, the drivers of inter-annual, seasonal, and diurnal variability of trace gases in
the atmosphere are part of a wide range of processes that include atmospheric dynamics and chemistry, but so far, a study of
the immediate influence of rain events is lacking.

Ozone concentration is strongly modulated by precipitation, as observed by Betts et al. (2002) during the Large-Scale Biosphere-
Atmosphere Experiment in Amazonia wet season experiment. They found an increase in ozone concentration and a decrease in
potential temperature as an indication of convective downdraft. They suggested the important role of the ozone increase in the
photochemical process in the boundary layer. Sigler et al. (2002) studied the effect of ozone on forested and deforested regions
in Amazonas. They found higher ozone concentrations over pasture than over forest, suggesting that the forest has a more
effective sink mechanism to consume ozone. Gerken et al. (2016) studied ozone dynamics during the GoAmazon experiment.
They compared concentrations between the dry and wet seasons and found that the average concentration differed by 5 ppbv,
higher during the dry season with an average concentration of 20 ppbv. They observed peaks of up to 25 ppbv in the boundary
layer during rain events. Lighting can also increase $O_3$ mixing ratios, as discussed by Shlanta and Moore (1972).

VOCs are reactive atmospheric trace gases and comprise many groups of saturated, unsaturated, and oxygenated derivatives
(Kesselmeier and Staudt, 1999). Biogenic VOCs (BVOC) include isoprenoids (isoprene and monoterpenes) as well as alkanes,
alkenes, carbonyls, alcohols, esters, ethers, and acids. Their reactivities are high, and their lifetimes in the atmosphere range
from < 1 min to days. Some species are hardly detectable under normal atmospheric conditions as they react too fast with
$O_3$ and radicals. Thus, BVOCs exhibit highly dynamic anomalies with strong diurnal and seasonal characteristics. For an
overview, see Kesselmeier and Staudt (1999) and Yanez-Serrano et al. (2020). Atmospheric mixing ratios strongly depend
on atmospheric oxidation processes, but the anthropogenic and biogenic production and emission pathways should never be
overseen. As reported by Laothawornkitkul et al. (2009), BVOCs are produced in the course of many plant physiological and
metabolic pathways involved in plant growth, development, reproduction, and defense. Their release to the atmosphere depends
on solubility and volatility and may, therefore, be a function of physiological gas exchange regulation under stomatal control.
Some of the BVOC species are released close to the mixing ratio gradient between outside air and plant tissue; some are
under strict stomatal control. This behavior strongly depends on water solubility, i.e., equilibrium gas–aqueous phase partition
coefficient (Niinemets, 2007). Thus, emission can occur in different manners, from constant diffusion to sudden bursts. Within
this context, we have to consider climate, season, and diurnal effects to include plant adaptation and development as well
as physiological reactions on shorter time scales, as slow and fast changes of BVOC emission in relation to adaptation and
developmental processes for plants, soil, and leaf litter. Within this context, bursts of gases and aerosols are often observed
during and after rain events (Greenberg et al., 2012; Bourtsoukidis et al., 2018; Rossabi et al., 2018). Such emission bursts, or

upward air mass transport, can be responsible for sudden changes in the hydroxyl radical (OH), an oxidative component acting as a sink for volatile organic compounds (VOCs) (Pfannerstill et al., 2021), and a catalyst for the further reactions of different trace gases in rain-forest, is strongly modulated by environmental variables, including rainfall, temperature, and radiation (Nölscher et al., 2016; Ringsdorf et al., 2023), and exhibits important vertical stratification and variations on intradiurnal to interannual scales.

This study aims to provide a comprehensive overview of the anomalies in trace gas mixing ratios due to rainfall events occurring within and immediately above the canopy at a site that serves as a representative sample of the central Amazon region. The analytical approach proposed in this study is important due to its capacity to unveil gas mixing ratio variability caused by rainfall. Accurately replicating these patterns by models is crucial for precise computation of the gas budget and life cycle. A composite study of the gas profile measurements conducted both during the day and night and inside and above the canopy provides a comprehensive opportunity to investigate the impact of rainfall on greenhouse gases (GHGs), specifically $CO_2$ and $CH_4$. Long-term vertical profile measurements enable us to analyze the interaction between soil, canopy, and boundary layer to specifically assess how the vertical profile of greenhouse and reactive gases vary before, during, and after rainfall events.

## 2 Data and Methodology

### 2.1 Measurement systems

This study utilizes the greenhouse gases (GHG) and reactive gases mixing ratio data collected at the Instant tower, a neighbor ATTO 80 m tower, at different heights with diverse instrumentation. For the gases NOx (nitric oxide (NO) and $NO_2$) and $O_3$, the data for this study was collected between 2013 and 2020 (for $O_3$) and between 2018 and 2020 (for NO and $NO_2$), and monitored mixing ratios at the heights 0.05, 0.5, 4, 12, 24, 38, 53, and 79 m, where 0.05 m hovers just above the surface, and 79 m is elevated approximately 45 m above the canopy. Each height was measured 4 times per hour, and air was sampled directly from the inlet height. Timestamps were rescaled to 30-minute intervals for each altitude. The mixing ratios of NOx (NO and $NO_2$) and $O_3$ were acquired using an Eco Physics CLD TR 780 and a Thermo Scientific 49i $O_3$ Analyzer, which measured by UV photometry with a precision of 1 ppb, as referenced in Andreae et al. (2015). Employing a gas-phase chemiluminescence technique, The CLD accurately measures the NO mixing ratio to an accuracy of better than 25 ppt. Subsequently, $NO_2$ was determined by converting it to NO through a photolytic converter driven by UV radiation (Solid-state Photolytic $NO_2$ Converter (BLC); DMT, Boulder/USA). Regular calibration was conducted with a Dynamic Gas Calibrator.

For CO, $CO_2$ and $CH_4$, the data for this study was collected between 2013 and 2020. Sample air from five different heights (4, 24, 38, 53, and 79 m) was led through a buffer system, such as described by Winderlich et al. (2010), which was connected to two instruments, employing cavity ring-down spectroscopy, the G1301 and G1302 analyzers (Picarro Inc.). The G1301 analyzer measures data with remarkable precision, displaying a minimal standard deviation of less than 0.05 ppm for $CO_2$ and 0.5 ppb for $CH_4$ in the raw readings. Furthermore, the device exhibits stability over time, with a long-term drift of under 2

ppm for $CO_2$ and 1 ppb annually for $CH_4$. The G1302 analyser, was calibrated using a stable gas tank. This comprehensive assessment revealed a standard deviation of 0.04 ppm for $CO_2$ and 7 ppb for CO in the raw data. Both instruments automatically measured 3 calibration gases every 100 hours, and a target tank every 30 hours. The measurement strategy was to cover all heights four times per hour, with a resampling rate of 30 minutes.

Isoprene ($C_5H_8$) and monoterpenes ($C_{10}H_{16}$) were collected at the Instant tower, using a Proton Transfer Reaction Mass Spectrometer (PTR-MS) from November 2012 to December 2015, but not continually due to the need of dedicated people for the instrument operation. Mixing ratios were measured at 0.05, 0.5, 4, 12, 24, 38, 53, and 79 m, where the canopy top is between 24 and 38 m. The sample inlets (3/8"OD insulated Teflon) were connected to the PTR-MS and installed at the foot of the Instant tower. Each level of the vertical profile was sampled every 2 minutes between different heights. This sequential operation allowed for a complete profile to be generated in just 16 minutes. The measurements were focused on two compounds: isoprene (m/z 69.069) and monoterpenes (m/z 137.132). For a detailed description, see Yáñez Serrano et al. (2015) and Yanez-Serrano et al. (2020). All trace gas profiles were linearly interpolated in 5-meter steps from the surface to 80 meters for better visual quality and equal vertical distribution.

Air temperature and relative humidity were measured at 26 m, wind speed at 42 m, and precipitation and solar radiation at the top of the tower at 79 m; all measurements were collected by weather sensors installed on the Instant tower. Temperature and relative humidity were measured using a Termo-hygrometer (CS215, Rotronic Measurement Solutions, UK), rainfall was obtained using a Raingauge (TB4, Hydrological Services Pty. Ltd., Australia), the wind speed was obtained through a 2-D sonic anemometer (WindSonic, Gill Instruments Ltd., UK) and the solar radiation with a Net radiometer (NR-LITE2, Kipp-Zonen, Netherlands). The data was collected from 2013 to 2020, but the parameters experienced intermittent failures at various times. From ABI (Advanced baseline imager) channel 13, total cloud cover was estimated, collocated at the ATTO site as the frequency of occurrence of brightness temperature ($T_{IR}$) $< 284$ K following Machado et al. (2021). The GLM (Geostationary Lightning Mapper) events, describing the lightning activity, were obtained from the GOES-16 GLM sensor, also collocated at ATTO site as the number of events every ten minutes in an area of 5 by 5 pixels in a 20 km radius, similar to Machado et al. (2021). $T_{IR}$ as well as GLM events were resampled every 30 minutes. Boundary layer heights were measured using a ceilometer model CHM15k (Jenoptik AG, Jena, Germany). The ceilometer is an instrument based on LIDAR, which involves capturing the intensity of optical backscatter in the wavelength range 900-1100 nm by emitting autonomous vertical pulses. LIDAR measurements are reliant on aerosol concentrations in the atmosphere. Within the PBL, aerosol concentrations are notably higher compared to the free atmosphere above, and this contrast serves as the foundation for detecting the PBL height through LIDAR measurements, see Dias-Júnior et al. (2022). Boundary layer height data were resampled every 30 minutes from 2014 to 2020.

$^{222}$Rn is a naturally occurring radioactive noble gas of terrestrial origin and is produced via the decay of long-lived radium isotope $^{226}$Ra, present in most rock and soil types (Nazaroff, 1992). $^{222}$Rn is measured in Bq.m$^{-3}$, corresponding to the amount

of radon radioactive decay per second in a volume of air and is used as a proxy of surface-atmospheric mixing and transport. $^{222}$Rn and $CO_2$ undergo similar exchange processes between the soil and the atmosphere, and all trace gases experience comparable atmospheric mixing phenomena as atmospheric mixing is turbulent. The exhalation of $^{222}$Rn from the soil can be considered almost constant in the absence of rain and changes in pressure. Consequently, it can serve as a proxy for assessing the dynamics of $CO_2$ emission/sink from/to the soil (Hirsch, 2007). Atmospheric radon activity concentration was measured at 80 m on the ATTO tall tower. Radon activity measurements covered the period from January 2019 to December 2020. The measurement used a static filter collecting radon progeny on a filter method and assuming radioactive equilibrium between atmospheric radon and its daughters as described by Levin et al. (2002), and a correction was made to account for the aerosol loss in the intake line as presented in Levin et al. (2017). The data was combined with rainfall to produce a composite for day and night to evaluate the relative importance of surface fluxes on the changes in the gas mixing ratio during rainfall events. The radon source can be assumed to be approximately constant over the diurnal cycle and horizontally uniform on local scales (Nazaroff, 1992). One limitation of this measurement is the error associated with situations where the relative humidity is larger than 95% or during rain events, when part of the atmospheric radon progeny may have been lost due to scavenging effects. In these situations, the general assumption that progeny are in equilibrium with the radon gas may be violated. To avoid this imprecision, the measurements used in this study account for only cases where the relative humidity is smaller than this threshold and are only applied for the range of two hours before the time of maximum rain.

## 2.2 Data analysis

We conducted a composite study following the same methodology as Machado et al. (2021), in which composites were based on the time of maximum rainfall events. This study intends to analyze the gas mixing ratio evolution during rainfall events by selecting moments within 4-hour time slots and using composite analysis to obtain the medium pattern evolution before and after maximum rainfall. Composite analyses are useful to study physical hypotheses that occur over time (Boschat et al., 2016). This method quantifies standardized instances of a specific phenomenon, such as a rainfall event, and consolidates them into a composite. In this study, we computed the gas mixing ratio during rainfall events with a maximum rain rate inside the four-hour time slot as the reference time. This analysis considered a time frame spanning two hours before and after the peak rain rate. A rainfall event was defined as any instance where the rain rate exceeded 0.5 mm.hr$^{-1}$ within a 4-hour window, with the peak occurring at the moment of maximum rain rate.

To compute the composites, we first define the median mixing ratio during the rain event as the median value of the trace gas $C$ at the height $z$ during all times when rain was observed during the day ($C_{day}(z)_{median}$), and the night ($C_{night}(z)_{median}$). The composite was constructed as the median at each time ($t$) corresponding to the window between 15 minutes before and 15 minutes after time $t$. The composite was performed for each trace gas, for day and night, for the time between 2 hours before and 2 hours after the maximum rainfall, every 30 minutes. The Mixing ratio composite difference is then calculated as:

$\Delta C_{day/night}(z, t) = C_{day/night}(z,t) - C_{day/night}(z)_{median})$. The composite was defined in this way to highlight the variability of the gas mixing ratio during the rainfall event.

We evaluated the composites at three distinct time points: at the onset, at the end of the rain event, and at the moment of peak rainfall intensity. The results exhibited qualitative similarity in gas mixing ratio evolution across these different calculations, except for the gas mixing ratio at the moment designated as the reference time. Supplementary Fig. S1 shows an example of the composite describing the ozone mixing ratio at a height of 79 m. It shows the variation during the rain event, at the onset, at the moment of maximum rainfall, and at the offset of the rain event. It can be seen that the composite does not show a very different behavior. The largest effect is at the moment of maximum rainfall because it fixes the time of maximum convective activity, the vertical transport, and the rainfall rate, which influence the turbulence and the rainfall. It can be seen that the definition of the moment as the zero time is not very different among the different composites; this is because a large part of the rainfall events (about 38% of the events) have a duration of less than half an hour. Therefore, the time of the beginning, the time of the maximum rain rate, and the time of the end of the rainfall event are considered simultaneously in the composite. The histogram of rainfall rate and event duration is discussed and presented in section 3.2. Considering that more than 64% of the rain events have a duration of less than 2 hours, the composite mainly comprises hours with and without rainfall. The main idea is to consider the mixing ratio variation before and after the maximum rain event rather than the absolute value or when the atmosphere returns to the background mixing ratio. We specifically chose the moment of maximum rain rate as the reference time to center the composite analysis on the most intense phase of rainfall activity. The selection of a 0.5 mm.hr$^{-1}$ threshold was based on its proximity to the resolution of the tipping amount. As it was expected that daytime and nighttime conditions are different, considering the presence or absence of solar radiation, we decided to separate the rain events into daytime events (occurring between 8:00 and 17:00 Local Manaus Time) and nighttime events (occurring between 20:00 and 5:00 Local Manaus Time). All composite results are presented as median values.

Between 2013 and 2020, a total of 1291 rain events were recorded. Since the collection period was different for each group of gases, the amount of to-be-studied rain events differed per group. The composite dataset utilized 647, 650, 673, and 774 rain events during the daytime and 286, 285, 291, and 264 rain events during the nighttime for $CO_2$, $CH_4$, CO, and $O_3$, respectively. For NO and $NO_2$, the data spanned the period from 2018 to 2020, but due to the intricacies associated with the measurement of NO and $NO_2$, certain time intervals experienced data gaps. Consequently, these gaps impacted the composite dataset when combined with rainfall data. The composite datasets involving rainfall events were derived from 114 and 54 rain events during the day and night, respectively, for NO. Similarly, for $NO_2$, the composites were constructed using 104 and 54 rain events during the day and night. In the case of Volatile Organic Compounds (VOCs), collected between November 2012 and December 2015, the composite datasets that considered rainfall events were calculated using 258 rain events during the daytime and 94 rain events during the nighttime. The various gases, their corresponding data collection periods, and the influence of rainfall events have contributed to a complex dataset that captures the dynamics of atmospheric constituents over the Central Amazon.

## 3 Results and Discussion

The analysis is based on a composite of several gas mixing ratio profiles covering GHGs and reactive gases, including selected
BVOC during rainfall events, as similarly studied for aerosol in the Amazon (Machado et al., 2021). This comprehensive
examination covers both daytime and nighttime periods. To enhance the discussion and interpretation, we commence the
analysis by examining the median profiles for each gas species during day and night. These profiles were computed inside the
4-hour time window, centered at the moment of the maximum rain rate. Subsequently, we will delve into the analysis of $^{222}$Rn
activity, utilized as a tracer for assessing surface-atmosphere mixing, alongside examining pertinent weather variables during
precipitation events.

### 3.1 Day and night mean profile of gas mixing ratio during rainfall events

The gas profiles shown in Figure 1 serve as the baseline for the composite analysis, highlighting variations in the profiles
around (4-hour window) the precipitation events. Composites were derived by calculating deviations from the median profile
within a 4-hour window, covering 2 hours before and 2 hours after the peak rain rate. The profiles presented in Figure 1
represent the basis for calculating gas mixing ratio differences in the composites. The canopy rises to heights around 32 m,
and as a reference, a line at 32 m was included in Figure 1. The Figures are presented as the median and the 30% and 70%
percentiles. The $CO_2$ profile exhibits distinct patterns characterized by elevated mixing ratios at night, a gradual reduction
within the canopy during the day, and a relatively constant mixing ratio above the canopy. During the nocturnal hours, the
vertical gradient of the mixing ratio decreases steadily. These profiles clearly show the nighttime source of $CO_2$ within the
canopy. Similarly, CO demonstrates analogous patterns, with its vertical distribution exhibiting similarities between day and
night. However, CO has a much larger variability. We should consider that these Figures include the dry and the wet seasons,
which have different background values. In contrast, $CH_4$ displays a similar mixing ratio during daytime and nighttime, and
its vertical variability remains minimal, typically less than 4 ppb. Even if the $CH_4$ medium is quite similar between day and
night and has a nearly zero vertical gradient, one can not the large variability between the 30%-70% percentile. Therefore, no
significant difference is found between day and night and in the vertical distribution. These profiles provide insights into the
different vertical dynamics of greenhouse gases with strong sources and sinks, like $CO_2$, and weak or absent local sources like
$CH_4$.

The $O_3$ profile follows a similar diurnal pattern, with higher mixing ratios during the daytime and an increase in mixing ratio
with height. $O_3$ mixing ratio and vertical variation are nearly identical during the day and night. $O_3$ varies from surface to 79
m by around 7 ppb during the day and 6 ppb at night. This day/night contrast becomes more pronounced above the canopy
level. These profiles illustrate how ozone-rich air infiltrates the canopy during rainfall events, contributing to its complexity.
The faster decrease in $O_3$ mixing ratios within the canopy primarily results from the interactions between BVOCs and $O_3$ and
reaction with the vegetation, as highlighted by Freire et al. (2017). Nevertheless, during instances of rain and, consequently,
downdrafts, a surge in turbulence occurs within the canopy, leading to the upward movement of air from the top to the bottom.

This atmospheric transport mechanism contributes to an augmentation in $O_3$ mixing ratios within the canopy, as elucidated by Mendonça et al. (2023). We also note a large variability of the $O_3$ due to the different background values during the dry and wet seasons (Supplementary Figs. S3). Conversely, the mixing ratio of NO remains primarily confined within the canopy, with marginal differences between day and night. However, a distinction arises just below the canopy, where nighttime mixing ratios are larger. This characteristic delineates the interplay of ground-level production and consumption processes within and

above the canopy, influenced by the dynamics of the nocturnal boundary layer and its impact on nighttime mixing ratio levels. $NO_2$ exhibits a distinct day-night contrast in its characteristics. At night, its mixing ratio remains relatively homogeneous, while during the day, there is an exponential decrease in the mixing ratio from within the canopy to higher altitudes. Additionally, nighttime mixing ratios above the canopy are higher, indicating production and accumulation within the canopy and consumption within and above during daylight hours.

Isoprene and monoterpenes display similar behavior, with their primary source at the canopy's top. Their mixing ratios decrease both above and below the canopy. During nighttime, isoprene exhibits minimal variation in mixing ratio with height, showing a nearly uniform distribution, with a gradual increase with altitude. In contrast, monoterpenes accumulate in the canopy at night and could be released during rain events. Although their mixing ratio is lower than during the day, their vertical distribution remains constant. The monoterpene production could be influenced by mechanical turbulence within the vegetation, especially

during rainfall events, and modulated by the air temperature.

Supplementary Figs. S2, S3, and S4 show the difference between the gas mixing ratio during rain and no rain events for day and night and wet and dry seasons. These figures provide information on the variability of the mixing ratio profile of each gas during the wet and dry seasons and the rainy and non-rainy moments, giving an idea of the background in each season and during the day and night. The $CO_2$ mixing ratio varies only about 2 ppm during the day and 8 ppm during the night between

265 rain and non-rain events. The variation between dry and wet seasons is small compared to other gases. For CO, the difference between rain and non-rain is around 2 and 6 ppb. Day and night variations are small compared to the seasonal variation of around 20 ppb. The variation in the rain and non-rain profiles have the same behavior between the seasons. The $CH_4$ variation due to rain and absence of rain is less than 1 ppb during the dry season and during the day. However, the largest vertical gradient is observed during the dry season and the day, with a maximum around the canopy. The difference between rain and the absence

of rain is about 3-4 ppb during the wet season and during the night. The seasonal difference is about 6 ppb. Supplementary Fig. S3 shows the ozone, nitric oxide, and nitrogen dioxide profiles. The $O_3$ has a significant difference between the dry and wet season mixing ratio; the day and night difference shows a different profile, with a more homogeneous profile within the canopy during the night. It is interesting to note that the non-rain event during the dry season has a higher ozone concentration, whereas the opposite behavior is observed during the wet season. This is because the background ozone mixing ratio during

the dry season is very high, and the wash-out effect seems to compensate for the effect of the injection of ozone-rich air from higher altitudes. NO shows a less pronounced seasonal variation. The profiles show a higher concentration near the surface and a faster decrease with height, and the non-rainy background shows a higher mixing ratio. $NO_2$ shows a nearly similar

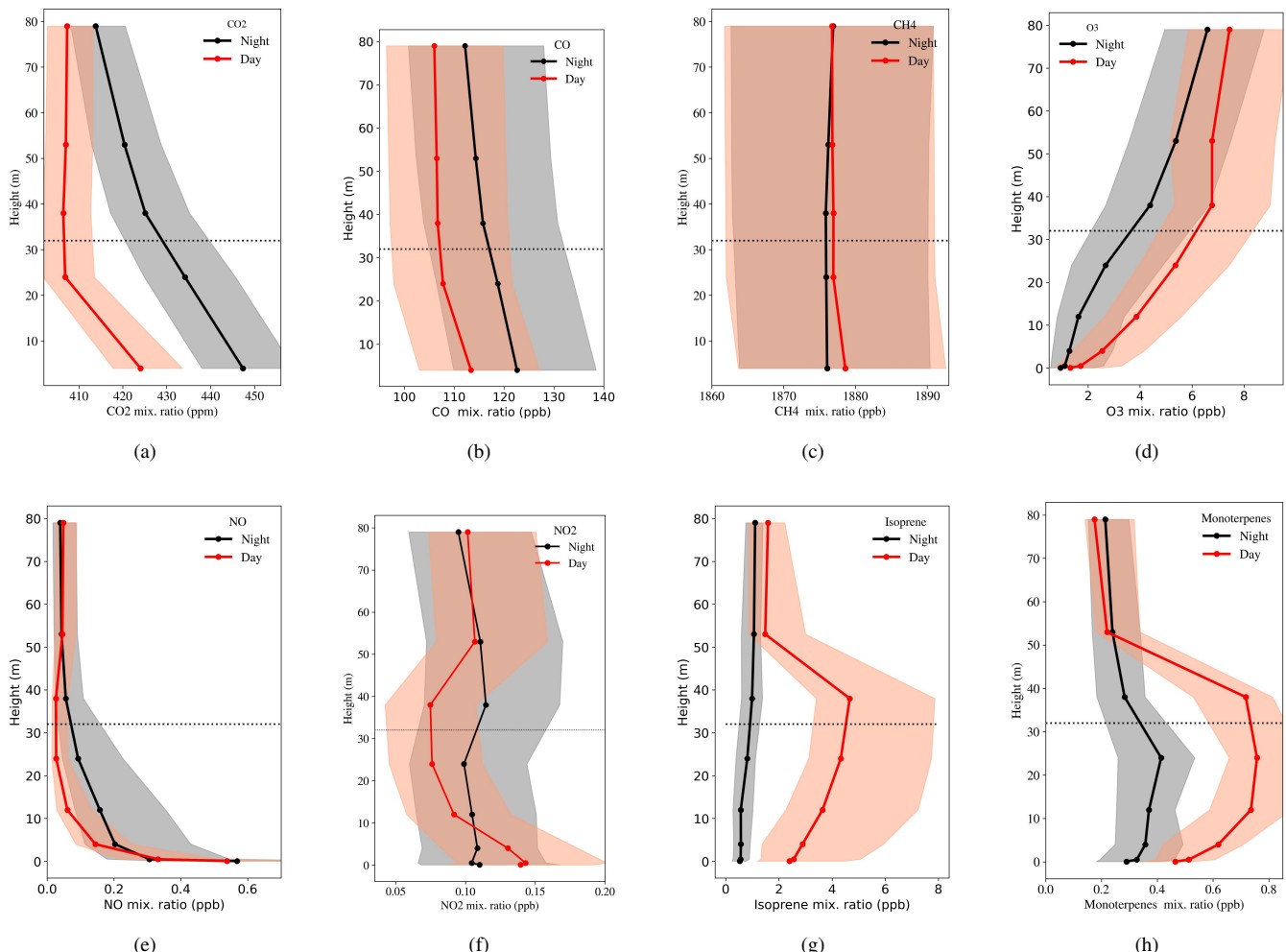

**Figure 1.** Vertical profiles of median mixing ratios and 30% and 70% percentiles interval (Shadow), for a) $CO_2$, b) CO, c) $CH_4$, d) $O_3$, e) NO, f) $NO_2$, g) isoprene, and h) monoterpenes during the day (red) and night (black), for all rain-events. The dataset to compute the median spans a two-hour window before and after the peak rain rate. The total number of rain events for each gas is detailed in the Data Analysis section.

variation with height during day and night, dry and wet, with a minimum mixing ratio just below the canopy, highlighting an important process of mixing ratio reduction in this height. Finally, Supplementary Fig. S4 shows the profiles of isoprene and monoterpene. All profiles show a maximum around the canopy and a different mixing ratio between rain and non-rain events. There is a considerable seasonal variation, with more than twice the concentration during the rainy season. Monoterpene has a similar or higher concentration during rain events than during non-rain events. Isoprene, however, has a higher concentration during non-rain events, except during the dry season, when the concentration is higher during rainy days.

The composite does not distinguish between the dry and wet seasons, even though most rain events occur during the rainy season. The main objective of this study is to evaluate the variation of the mixing ratio of a specific gas with the rain event, observing before and after the maximum rain rate. Therefore, a composite was made from two hours before to two hours after the maximum rain rate, and the average of the median profile during these four hours was calculated (Figure 1) and subtracted from each time step. This was done to highlight the variation during the rain event rather than the absolute concentration, which varies between days and seasons. Therefore, the mean values of the composite should be zero.

## 3.2 Vertical mixing and environmental characteristics during rainfall events

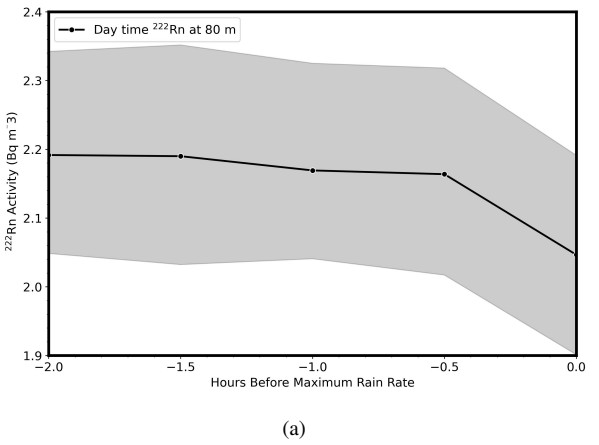
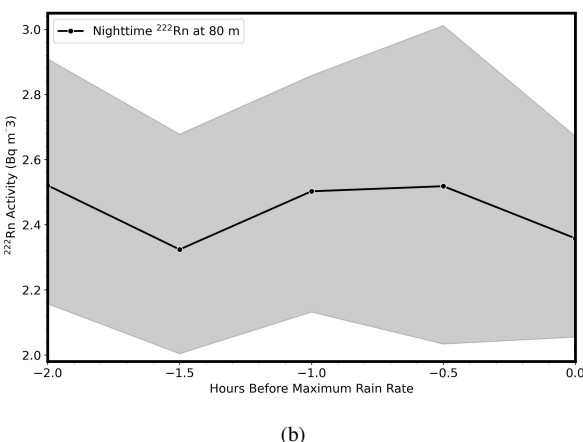

(a)                                                             (b)

**Figure 2.** Composite of $^{222}$Rn concentrations at 80 m above ground during day (a), and night (b), from two hours before to the time of the maximum rain rate. After the rain, part of the atmospheric radon progeny could have been lost due to scavenging effects and to avoid this imprecision $^{222}$Rn activity concentrations are not reported after maximum rain rate. A rainfall event was considered as an event inside the 4-hour window with at least one moment with a rain rate larger than 0.5 mm.hr$^{-1}$. The data used covers rain events from January 2019 to December 2020. Values are presented as the median values, and the gray shading is the 95% confidence interval.

Changes in gas mixing ratio during rainfall events could be modulated by an ensemble of effects, from the vertical advection of free troposphere air, changes in surface fluxes, changes in the cloud cover inducing a reduction in solar radiation reaching

the canopy, temperature changes, the wash-out effect by the rain, among others processes Pedruzo-Bagazgoitia et al. (2023). We used a proxy value based on atmospheric $^{222}$Rn activity concentration measurements to evaluate the effect and dynamics of the soil fluxes. Of course, this proxy only applies to gases that are co-emitted with $^{222}$Rn from the soil, such as $CO_2$, NO and possibly $CH_4$ after a rain event. The relative humidity threshold (<95%) employed in the data analysis considerably reduced the sample size; for the day events, the composite had 109 cases, and for the night, 23 cases. Figure 2 shows how $^{222}$Rn varies before the maximum rainfall event during the day and during the night, measured at 80 m height above the ground. During the day, the $^{222}$Rn activity concentration is nearly constant 2 hours before the maximum rain rate. There is a slight decrease half an hour before the moment of maximum rainfall, which is likely due to the effect of the onset of precipitation, causing washout and radioactive imbalance thereafter. On the other hand, during nighttime, radon concentrations peak one hour before the maximum rain intensity, followed by a slight drop in activity concentration and also half an hour before the moment of maximum rainfall, possibly due to the same wash-out effect. We should observe a large variability during the night due to the small sample; the large variability of the sampling and the wide confidence interval do not allow us to say that these changes in behavior associated with rainfall are statistically significant. However, we tested the behavior separately for only 2019 and 2020, and the pattern of maximum activity concentration before maximum rainfall is consistent between the two years. This analysis suggests that preceding rain events, there is a nearly constant surface flow during the day and an increase in surface flows at night. These features will be discussed in detail in the upcoming subsections. The increase in radon activity concentrations is more pronounced during nighttime, possibly linked to atmospheric stability leading to the accumulation of trace gases with a local source. For further evaluation of the $^{222}$Rn behavior with other gases ($CO_2$, CO, $CH_4$) see Supplement Figure S5 showing the behavior side by side, with normalized variation based in standard variation. These Figures will be discussed during the analysis of each of these gases in the next session.

Figure 3a) and b) show the histogram of the duration of the rainfall events and the maximum observed rain rate of each event. Most of the events have a duration of less than an hour and a rain rate of less than 1 mm.hr$^{-1}$; therefore, the composites are mainly composed of short events with low rain rates. There are some cases with high rain rates and event duration of more than 400 minutes and 25 mm.hr$^{-1}$, but about 64% of the cases have a duration of less than 2 hours and rain rates of less than 5mm.hr$^{-1}$. The maximum rain rate could occur at any moment of the rainfall event. However, it currently occurs closer to the rainfall initiation rather than the end of the event due to the stratiform rain that follows the convective, most intense, rain. Figure 3c) illustrates the temporal variations in several key meteorological parameters, including temperature, relative humidity, wind speed, solar radiation, boundary layer height, and lightning events within a 4-hour window surrounding the time of maximum rain rate. Temperature exhibits a decreasing trend from 1 hour prior to the peak rainfall, reaching its lowest point approximately 30 minutes after the maximum rainfall intensity. Conversely, relative humidity experiences a steady increase, with the highest humidity levels occurring about 30 minutes after the peak rainfall. Wind speed attains its maximum value at the time of the highest rainfall rate, while total cloud cover shows an overcast situation 1.5 hours before the maximum rain rate, and solar radiation registers its lowest value concurrently with the maximum rain rate. Boundary layer height initiates a decline around

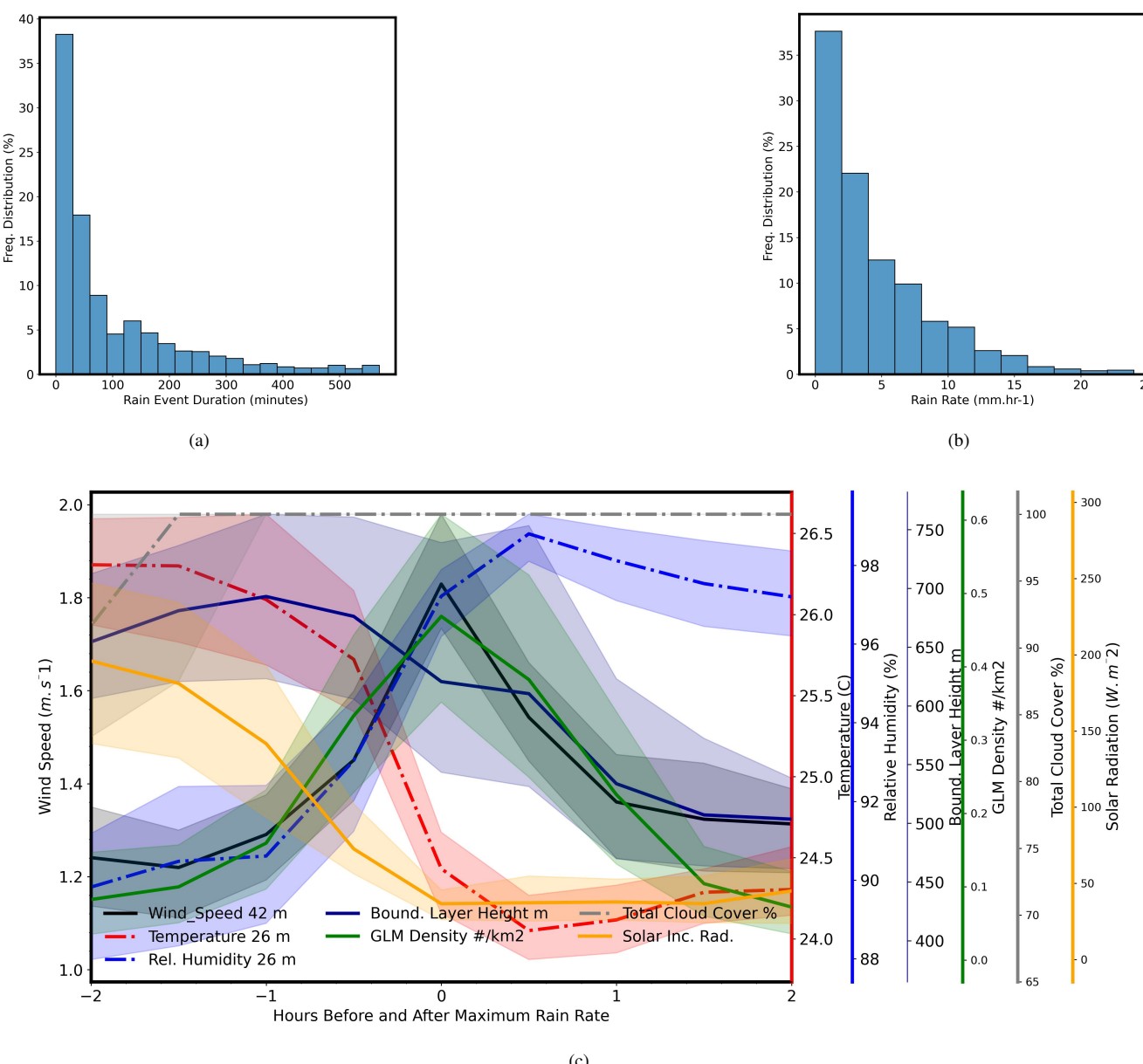

**Figure 3.** a) Rain event duration histogram for all cases in the composite, b) Maximum rain rate distribution for all cases in the composite, c) Composite of weather variables for two hours before the maximum rain rate to two hours after. The air temperature was measured at the height of 26 m, relative humidity at the height of 26 m, wind speed at the height of 42 m, and solar radiation at the top of the tower at the height of 79 m. The Figure also includes the GLM lighting events, the boundary layer height, and the cloud cover. A rainfall event was considered as an event inside the 4-hour window with at least one moment with a rain rate larger than 0.5 mm.hr$^{-1}$. Values are presented as the median values, except for the GLM lighting, because lighting events are not frequent and the median value is zero, so we used the mean for GLM lighting. The shaded bands around the median or mean values correspond to the 95% confidence interval.

30 minutes before the maximum rainfall, indicating a notable influence of precipitation on the atmospheric boundary layer. Furthermore, lightning events (hereafter called GLM events) reach their maximum simultaneously.

Rainfall is correlated with changes in other meteorological variables, as described above, but gas solubility is also directly affected by rain events. Rain can increase the rate of air-water gas exchange, Ho et al. (1997) empirically calculated the gas transfer velocity and rainfall rate for different rainfall rates and drop sizes, quantifying the enhancement of air-water gas exchange by rainfall. There is also the wet deposition effect, which may not be an efficient removal mechanism for hydrophobic gases as described by Mullaugh et al. (2015). The direct rainfall effect depends on the solubility of gases, and wet deposition is highly complex, especially for VOCs, due to the water solubility of this heterogeneous mixture covering several orders of magnitude (Niinemets and Reichstein, 2003).

## 3.3  Greenhouse gases, carbon monoxide and rainfall events

To effectively illustrate the variations in the gas profiles before and during precipitation events, the calculations were performed as deviations from the average profile within 4-hour windows, spanning 2 hours before and 2 hours after the peak rain rate. To enhance visual clarity, the figures represent the deviation inside the 4-hour window, centered in the moment of maximum rain rate, and employing distinct scales to accentuate pre- and post-rain contrasts. The composite is used to describe the changes in the gas mixing ratio with the rain event. We calculated the diurnal cycles of the gas mixing ratio with and without rain to demonstrate the significance of the differences between the gas concentration during the rainy and no-rainy events. Supplementary Fig. S6 shows the median of all gas species mixing ratios for the rainy and no-rainy events. The figures show the quantitative difference between the two situations and provide an indication of how a particular gas concentration varies during rain events. The discussion of each gas and its evolution during the rain events is discussed in detail in the following figures.

Figure 4 depicts the diurnal and nocturnal evolution of the deviation from the median profile, presented in the precedent section, for $CO_2$, CO, and $CH_4$ mixing ratio profiles within a two-hour window before and after the peak of rainfall events.

The majority of gas mixing ratios exhibit a decline concurrent with precipitation. The $CO_2$ profile varies between the surface and above the canopy at 79m, before and after the rain event, around 1.8 ppm during the day to 3.3 ppm at night (total range of variation). While $CO_2$ mixing ratio shows limited sensitivity to rain, particularly during the day, the influence is more pronounced at night, manifesting a significant reduction in mixing ratios below the canopy. This data underscores the variable response of $CO_2$ mixing ratios to rainfall, emphasizing its more prominent role in nocturnal conditions, possibly due to the enhanced mixing conditions associated with rainfall having a relatively larger impact on the nighttime $CO_2$ mixing ratio buildup than during the daytime. $CO_2$ mixing ratios, possibly affected by local sinks and sources (vegetation and soil), behave differently during day and night. During the night, the largest variation is below the canopy but in phase with the heights above.

During the day, there is a decrease in concentration inside the canopy with the rainfall, but above the canopy, the variation is very small with a slight increase with rainfall. Probably because of the reduction of photosynthesis due to the increase in cloud cover, it could result in a slight increase in $CO_2$ mixing ratio above the canopy. Higher mixing ratios near the ground and lower ones at the canopy height suggest sources close to the forest floor, such as soil and understory ecosystem respiration, and a stronger photosynthetic sink at the higher heights. The decrease in the $CO_2$ mixing ratio within the canopy during the rain event is correlated with the simultaneous increase in humidity and decrease in temperature as a consequence of the reduction in radiation due to the increase in cloud cover Pedruzo-Bagazgoitia et al. (2023). As discussed above, these environmental conditions suppress both soil and tree $CO_2$ exchange and surface flux and reduce photosynthesis. Another possible reason could be associated with increased mixing within the canopy, destroying the stable layer within the canopy by mixing free tropospheric air into the canopy Betts et al. (2002). These two effects may contribute to the reduction in $CO_2$ mixing ratio after the rain event; however, the importance of each of these effects could not be quantified with the current data.

During the night before the rain event, there is a clear increase in the mixing ratio at 79 m according to the radon surface flux proxy (see Supplementary Figures S5). The nocturnal production of $CO_2$ combined with the turbulent fluxes associated with the gust fronts of the rain events may increase the mixing ratio above the canopy. The rain event leads to turbulent air mixing from above down to the ground, resulting in a strong decrease of $CO_2$ mixing ratio.

The evaluation of CO mixing ratios around rain events (Figure 4c,d) shows a similar behavior during the day and night, though with some important differences. Before the rain event, CO profiles exhibit high differences near the ground and show lower mixing ratio differences near the canopy, mainly during the day. Thus, there is likely a source of CO near the forest floor (van Asperen et al., 2023). In global CO inventories, the biosphere is regarded to act as both a source and a sink, but large uncertainties remain about the strength of individual sources. CO emissions are usually associated with abiotic degradation of organic matter, in the form of photodegradation (Guenther, 2002; Seiler and Conrad, 1987; Schade et al., 1999; Tarr et al., 1995; Derendorp et al., 2011) as well as thermal degradation (Yonemura et al., 1999; Lee et al., 2012; van Asperen et al., 2015). Living plants have also been reported to show CO emissions, but are expected to be minimal compared to senescent plant material (Derendorp et al., 2011; Schade et al., 1999; Tarr et al., 1995). Besides soil CO emissions, soil CO consumption cannot be excluded: soil microorganisms are known to oxidize CO to $CO_2$, a process, among others, dependent on available oxygen (soil diffusivity) and temperature (King and Hungria, 2002). As underlined by Liu et al. (2018), the balance between soil CO uptake and soil CO emission is not well understood, especially in the tropics. These effects are more important during the wet season because the anthropogenic effect (e.g. fire emissions) is very significant during the dry season. As for the $CO_2$ profile, during the night, the rainfall effect is more vertically homogeneous.

During the night, CO shows a slight vertical gradient with higher CO mixing ratios difference close to the forest floor. The precipitation event causes a more important difference near the ground and, may be associated with the processes of uptake and emission of CO with the same transport phenomena discussed for $CO_2$. Above the canopy, at night, at 79 m, an increase

in the mixing rate of CO after rain is not observed (see Supplementary Figure S5), related to the increase in surface turbulent fluxes, as observed for $CO_2$, but less clear for a probably smaller nocturnal source of CO, comparable to that of $CO_2$ (see Supplementary Figure S5).

$CH_4$ (Figure 4e, f), in contrast to the other gases, displays notably less stratification, with variations spanning from the surface to 79 meters, amounting to less than 4-6 ppb throughout both day and night periods (see Figure 1c). Although a slight stratification can be observed in this figure, a discernible pattern emerges with the highest mixing ratio differences occurring at the surface during the day, while at night, they are more prevalent at higher levels above the canopy. This daytime behavior can be attributed to weak sources of $CH_4$ production, primarily stemming from microbial anaerobic decomposition processes, depending on temperature and soil humidity, occurring mainly near the ground. As temperatures decrease during nighttime, $CH_4$ production wanes, possibly leading to the observed shift in mixing ratio peaks toward upper levels. Assuming the inlet at 79-m height is within the nocturnal boundary layer, the nocturnal maxima can be explained by the processes described in Botía et al. (2020), but if the inlet height is above the nocturnal boundary layer and inside the residual layer, the $CH_4$ peak could be associated with the mixing ratio of the previous afternoon. As $CO_2$ and CO, $CH_4$ exhibits a time trend with rainfall with a minimum mixing ratio difference of two hours subsequent to the peak rainfall intensity. The supplementary information presented in Figure Supplement 5 shows a practically unchanged mixing ratio of $CH_4$ during the night before the rain events, which indicates that although there are turbulent fluxes at the surface, the mixing ratio is not affected above the canopy, possibly due to the absence of sources at that time.

Generally, the greenhouse gas mixing ratios ($CO_2$, CO, and $CH_4$) revealed a similar pattern during day and night and, overall, there is a noticeable declining trend in their mixing ratios after reaching the peak of the rain intensity. This pattern implies that atmospheric transport plays a pivotal role in regulating the levels of trace gases, as evidenced by the concurrent rise in wind speed and boundary layer height, as depicted in Figure 3, leading up to the maximum rain rate.

During the nighttime, the influence of atmospheric transport on $CO_2$, CO and $CH_4$ mixing ratios difference before reaching the maximum rain rate is less stratified compared to daytime. The mixing ratios of these gases exhibit a more homogeneous vertical variability until the point of maximum rain intensity. Still, a more intense effect on the mixing ratio difference for $CH_4$ is observed near 79 m.

The effect of air transport from the free troposphere to the canopy should be the same for all the gases, as it is related to the amount of air exchanged. However, the source-sink patterns of the three gases in terms of time and location within the canopy differ, implying the pattern observed in these compounds. The data used in this study does not allow transport effects to be completely separated from the effects of sources and sinks. The composite highlights distinct patterns associated with the rain events in the gases presented here. For $CO_2$, the strong vertical gradient for night and day with high mole fractions within the canopy suggests a local production of the gas and poor $CO_2$ gas mixing ratio transported from the free atmosphere. In contrast,

CO profiles could be affected by vertical transport, whereas the minimal vertical gradient of CH₄ could indicate an almost
balanced local production/sink relationship.

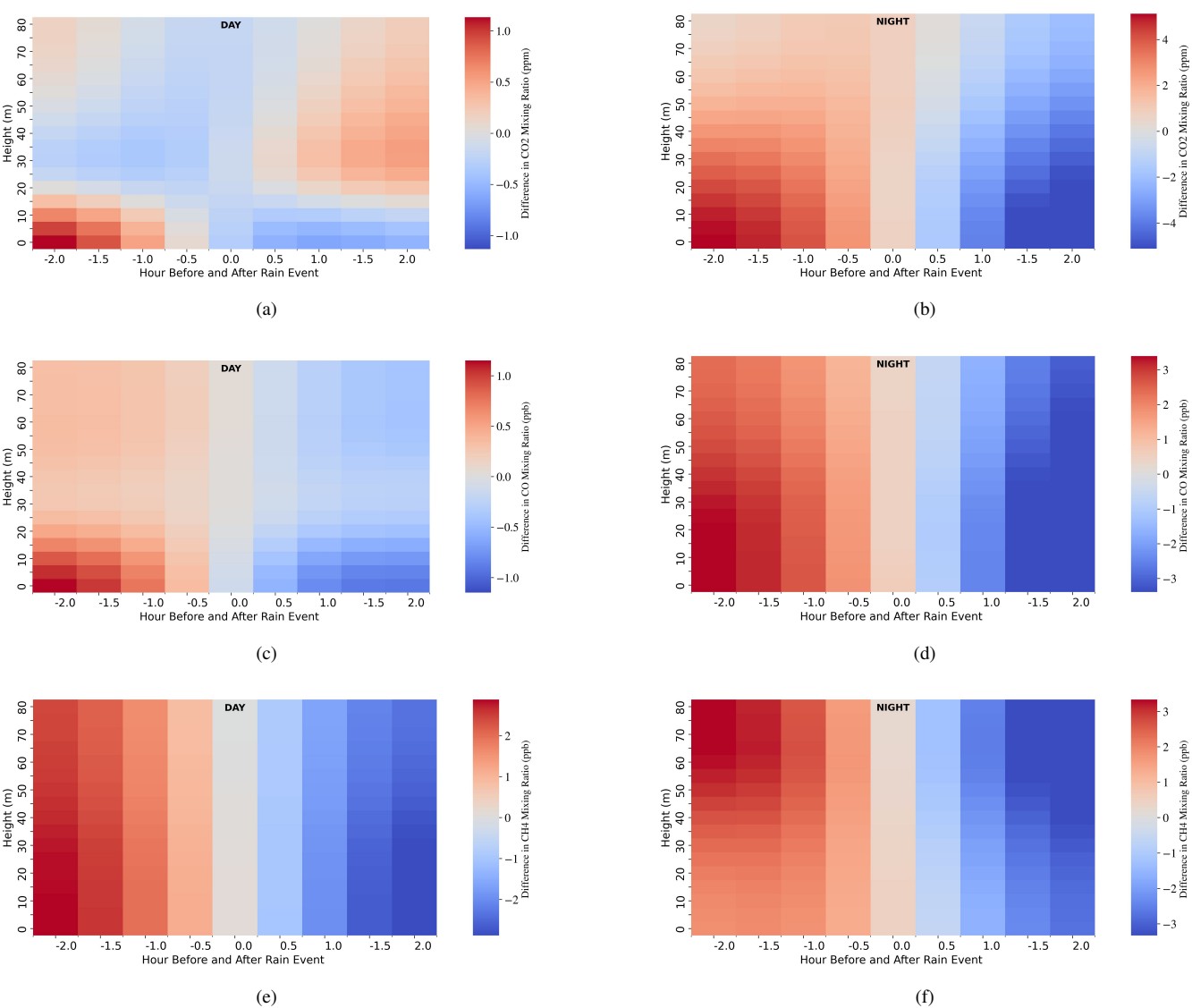

**Figure 4.** Composite of CO₂ during the day (a), and night (b), CO during the day (c) and during the night (d), and CH₄ during the day (e) and during the night (f), for two hours before the maximum rain rate to two hours after. Please note that the color bar scale is different between each figure. A rainfall event was considered as an event with at least one case with a rain rate larger than 0.5 mm.hr⁻¹. Calculation was done for all cases from January 2013 to December 2020. Values are presented as the deviation from median composite values.

## 3.4 Reactive gases and rainfall events

Figure 5 shows the evolution of $O_3$, NO, and $NO_2$ mixing ratio profiles during the day (a,c,e) and during the night (b,d,f), two hours before and after the maximum rainfall. Figure 5 (a,b,) shows $O_3$ mixing ratio increasing during precipitation events, as observed by Wang et al. (2016); Gerken et al. (2016); Sigler et al. (2002). A considerable increase is driven by a downdraft reaching the ground. This injection of ozone-rich air from the upper levels reaches a maximum of about one hour after rainfall. The $O_3$ increase within the canopy can be attributed to the injection of high levels of $O_3$ mixing ratio from the upper troposphere. $O_3$ mixing ratio is related to atmospheric chemistry, cloud dynamic transport, and cloud electrification (Brune et al., 2021; Williams et al., 2002), as indicated by a maximum lightning activity at this time. Of special interest is the less striking variation of the $O_3$ mixing ratio just around and above the canopy during the day, before and after rainfall. This result suggests deposition, decomposition, or uptake by vegetation. This kind of sink masks the variation as affected by the rain event. As this level represents the main source for isoprene and monoterpenes, mainly during the day, a reaction with these VOC species may play a crucial role. The $NO/NO_2$ dynamics within the precipitation event seem to reflect this complex series of reactions near the surface and above the canopy, starting with soil NO emission and accumulation affected by $O_3$, transport processes and chemical reactions, resulting in the production of $NO_2$. This general view is supported by several reports based on chamber and field experiments (Rummel et al., 2002; Gut et al., 2002; Kesselmeier et al., 2002; Chaparro-Suarez et al., 2011; Bell et al., 2022; Zhao et al., 2021) indicating the oxidative regime is governed by $O_3$ and affecting several trace gases. These studies contribute to understanding the within-forest oxidative capacity reflected by VOC oxidation products, such as formaldehyde, as observed under daytime conditions near the forest surface (Rottenberger et al., 2004). During the day, the rainfall event leads to an opposite behavior between $NO_2$ and NO; the former is increased during rainfall, and NO decreases after the rainfall. NO variation with rainfall mostly occurs within the canopy. During the night, the effect is also observed above the canopy, possibly due to the small height of the nocturnal boundary layer. For $NO_2$, the variation with rain is highest just above the canopy. However, at night, a different pattern emerges, showing the importance of solar radiation in the daily photochemical reaction. At night, the highest variations of $NO_2$ mixing ratio primarily occur around the canopy top. This can be understood as caused by vegetation's missing $NO_2$ uptake. Under daylight, with open stomata, the plant leaves are effectively taking up $NO_2$ from the air, whereas stomata are closed in the dark, and the sink strength decreases to negligible (Chaparro-Suarez et al., 2011). The rainfall events at night show $NO_2$ being washed out from the air above the canopy, with the strongest loss at the canopy top. Interestingly, a small positive variation remains near the ground after the rainfall. However, this potential source remains unclear as the rain does not activate any soil NO source to produce $NO_2$, but could be the product of the NO and $O_3$ reaction producing $NO_2$. Nevertheless, the behavior of NO anomalies within the forest shows an expected picture of accumulation before and decreasing mixing ratio after rain. Although the absolute difference with rain events is small, it corresponds to about 10% variation. The NO variation with rain is restricted to the inside canopy during the day and the night.

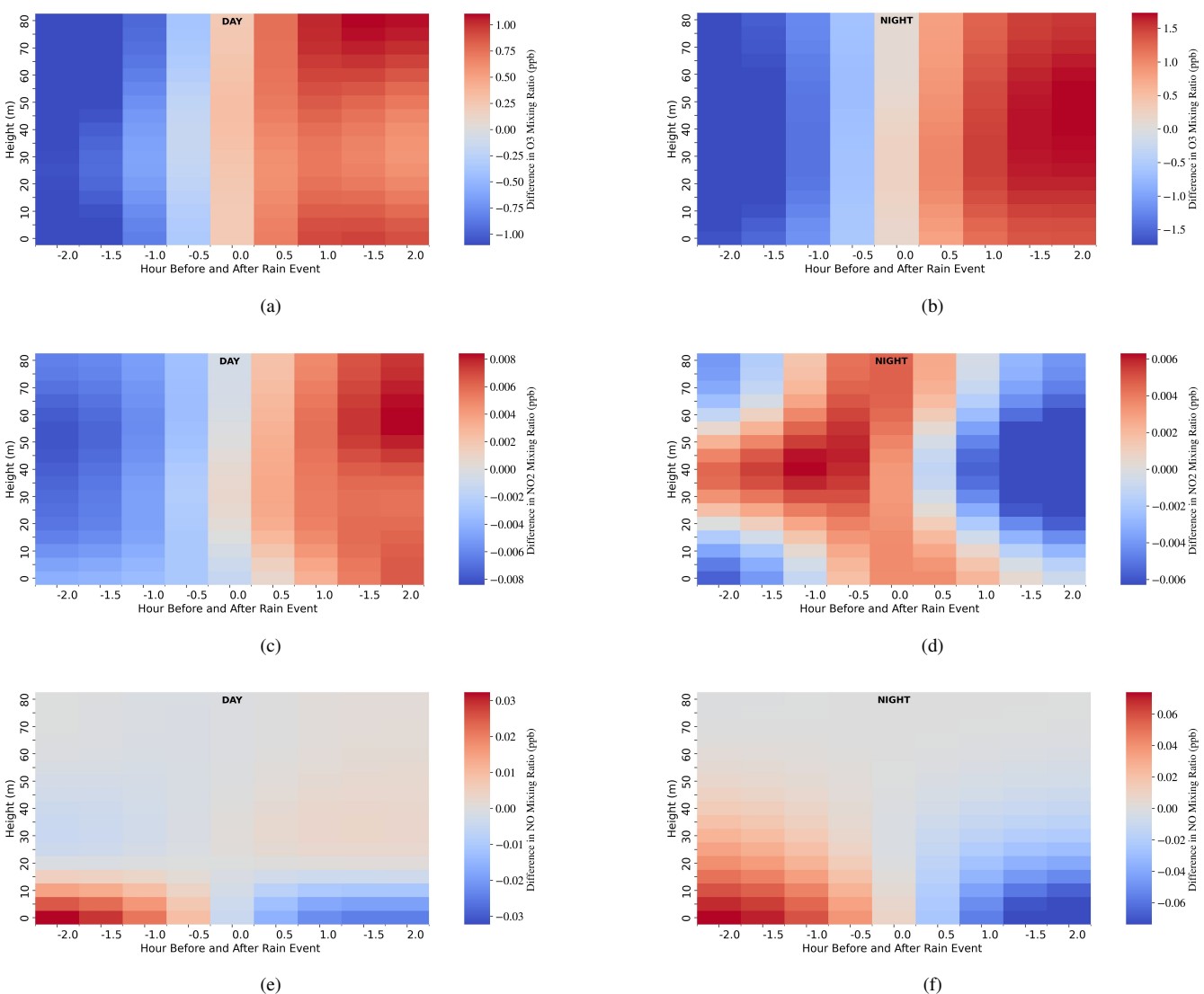

**Figure 5.** Composite of O₃ during the day (a), and night (b), NO₂ during the day (c) and during the night (d), and NO during the day (e) and during the night (f), for two hours before the maximum rain rate to two hours after. A rainfall event was considered as an event inside the 4-hour window with at least one moment with a rain rate larger than 0.5 mm.hr$^{-1}$. Calculation was done for all cases from January 2013 to December 2020. Values are presented as the deviation from median composite values.

## 3.5 Volatile Organic Compounds and rainfall events

Figure 6 shows the evolution of isoprene and monoterpenes mixing ratio profiles during the day (a,c) and during the night (b,d) two hours before and after the maximum rainfall. The depletion of biogenic VOC under these conditions could generally be related to oxidative atmospheric chemistry leading to secondary organic aerosol. The most important biogenic non-methane VOCs, isoprene and monoterpenes, have their maximum mixing ratio difference increase about two hours before the rain peak during the day related to canopy release and upward transport, driven by the meteorological conditions before rainfall, mostly with high air temperatures and intensive solar radiation favoring the biogenic synthesis of isoprene and monoterpenes. Typically, when rain starts in a forest, the temperature and solar irradiance drop rapidly, directly affecting biogenic trace gas production. Subsequently, during this time (daytime), the mixing ratios continue to be lower than before the rain event. This basic behavior is observed at all levels and is comparable to the observations of isoprene after rain events reported in Pfannerstill et al. (2021). The clear loss of isoprene and monoterpene emission causes a sharp mixing ratio decrease above the canopy. This drop indicates a set of decomposition pathways, deposition processes, and consumption inside the canopy. Under night conditions, a constant decrease of isoprene is observed before the rain event, followed by a slight increase of isoprene near the ground under rainfall, and it even affects higher heights in the course of time after rainfall. This observation of a slight increase in the isoprene mixing ratio at night looks puzzling, but only at first view. We have to regard the biosynthesis of this compound. Plants produce isoprene with chloroplasts (see Lichtenthaler (1999)) for an overview. But the chloroplastic pathway is of bacterial origin (Rohmer et al., 1993, 1996) imported by endosymbiosis, a relic of bacterial pathways through the evolution of plastids, obviously. More recently, bacterial isoprene synthesis was confirmed and described as a general issue of the bacterial genome (Sivy et al., 2002; Rudolf et al., 2021). Thus, we understand the isoprene increase and accumulation near the ground under night conditions due to a light-independent microbial production of this compound, which becomes visible under the applied composite description around rainfall. Furthermore, this pathway might also contribute to further isoprenoid accumulation, i.e., that of monoterpenes. Monoterpenes have a lower mixing ratio during the night, but in contrast to isoprene, the pattern caused by the rain event shows a clear accumulation before the rainfall peaks. This maximum is located at a lower altitude above the forest canopy than during the day. Obviously, there is a source within the lower layers of the forest. We may only speculate about such potential sources, for example, plant tissues with stored monoterpenes physically affected by raindrops, such as glands or hairs, or mechanical turbulence caused by the gust front associated with the convective process.

## 4 Conclusions

The combination of rainfall data with gas profiles collected at the ATTO site over several years provides insight into the trace gas variability during rainfall events. This analysis provides a quantitative description of greenhouse gases, reactive inorganic and volatile organic compounds at different heights within and above the canopy and being affected by meteorological events.

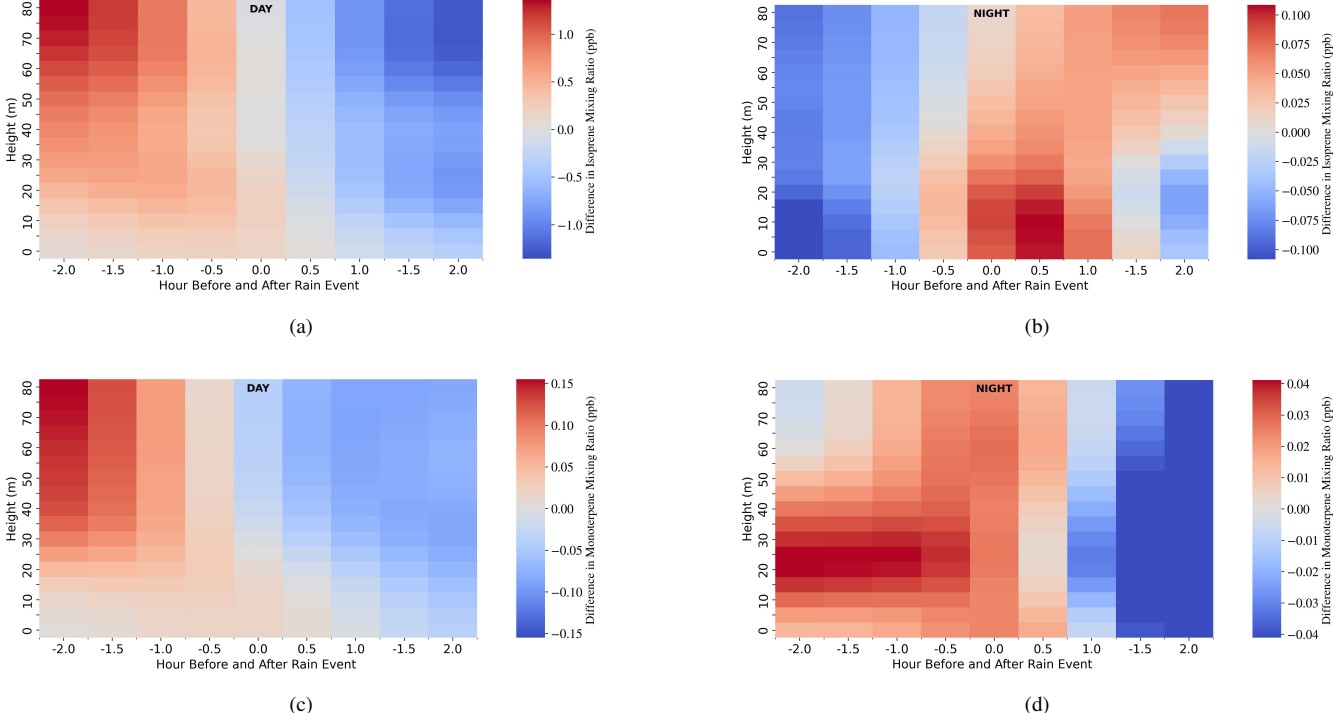

**Figure 6.** Composite of isoprene during the day (a) and night (b), monoterpenes during the day (c) and during the night (d), for two hours before the maximum rain rate to two hours after. A rainfall event was considered inside the 4-hour window with at least one moment with a rain rate larger than 0.5 mm.hr$^{-1}$. Calculation was done for all cases from January 2012 to December 2015. Values are presented as the deviation from median composite values.

The average profile for each gas computed based on the median-composite 2 hours before to 2 hours after a rainfall event during day and night clarifies several aspects of the gas behavior within and above the canopy. BVOC, $O_3$ and $CH_4$ present larger mixing ratios during the day, and $CO_2$, CO, and NO have maximum mixing ratios during the night, $NO_2$ has a mixed behavior, with a larger mixing ratio near the ground during the day and above the canopy mixing ratio is larger during the night. For all species, except for $NO_2$, the nighttime profile is more homogeneous due to the shallow nocturnal boundary layer. These profiles suggest the source of each gas, such as the canopy top for VOC, respired from leaves and stems and ground surface for $CO_2$, and ground surface for CO, and $CH_4$ and free-atmosphere for $O_3$.

The $^{222}$Rn is used as a surface flow tracer before rain events to complement the data analysis. There is a difference between day and night. During the day, the activity of $^{222}$Rn remains practically unchanged for two hours before the moment of maximum rainfall, while at night, there is an increase in the mixing ratio of radon activity one hour before the moment of maximum rainfall. This indicates a greater impact on the soil surface flux during the nocturnal rain event.

Composite analysis of the gas mixing ratio before and after rainfall, during the day and night, gives insight into the complex relationship between trace gas variability and precipitation. Entrainment from above may affect the mixing above and within the forest. This analysis improves our understanding of trace gas sinks and sources. $CO_2$, CO, and $CH_4$ decrease with rain, probably related to the clean air injected into the boundary layer from the upper levels. $CO_2$ and monoxide are more stratified with height than $CH_4$. CO has a sharper change with rainfall than $CO_2$. $CH_4$ changes are less significant, proportionally to the total mixing ratio, than $CO_2$. $O_3$ mixing ratio difference increases during precipitation events. The variability patterns of the NOx family in time and space are closely related to the contributing sink and source processes. As discussed above, a series of potential interactions exist between soil, atmosphere, and plants inside the canopy. Biogenic VOCs, such as isoprene and monoterpenes, change with the rainfall affected by light (production) and physical transport. The NO-$NO_2$ emission and reaction chain becomes visible concerning soil emission of NO, resulting in an accumulation at night or oxidation to $NO_2$ and release from the forest during the day. Furthermore, rainfall can activate trace gases (NO) burst from soil or VOCs from plant storage tissues. Thus, this composite analysis helps to understand sources and sinks of trace gases within a forest ecosystem.

*Data availability.* All data used in the study will be available at the Edmond repository, with open access.

*Author contributions.* Conceptualization: LATM, CP

Data curation: HVA, SPJ, JL, AYS, VIDV, SK, DW, SW, IL

Funding acquisition and administration: UP, PA, JL, CAQ, CP, ST, IL

Methodology: LATM

Writing original draft: LATM, JK, SB

Writing review & editing: All authors

*Competing interests.* The contact author has declared that none of the authors has any competing interests

*Acknowledgements.* This research has been supported by the Max Planck Society, the FAPESP grant 2022/07974-0, and the Bundesministerium für Bildung und Forschung (BMBF contracts 01LB1001A, 01LK1602B, 01LK1602C and 01LK2101B). For the operation of the ATTO site, we acknowledge the support of the Max Planck Society, the German Federal Ministry of Education and Research, and the Brazilian Ministério da Ciência, Tecnologia e Inovação as well as the Amazon State University (UEA), FAPEAM, LBA/INPA, and SDS/CEUC/RDS-Uatumã. We would like to especially thank all the people involved in the technical, logistical and scientific support of the ATTO project, in particular Reiner Ditz, Hermes Braga Xavier, Nagib Alberto de Castro Souza, Adir Vasconcelos Brandão, Valmir Ferreira de Lima, Antonio Huxley Melo do Nascimento, Amauri Rodriguês Perreira, Wallace Rabelo Costa, André Luiz Matos, Davirley Gomes

Silva, Fábio Jorge, Thomas Disper, Torsten Helmer, Steffen Schmidt, Uwe Schultz, Karl Kübler, Olaf Kolle, Martin Hertel, Kerstin Hippler, Fábio Jorge, Björn Nillius, Thomas Klimach, Delano Campos, Juarez Viegas, Sipko Bulthuis, Fernando Goncalves Morais, Roberta Pereira de Souza, Bruno Takeshi and Francisco Alcinei. We thank the editor and reviewers of our manuscript for providing constructive feedback to improve it.

**References**

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
