# Peer review of "How Rainfall Events Modify Trace Gas Mixing Ratios in Central Amazonia"

_EGUsphere, 2023_

## Referee Comment (RC1)

Review of "How Rainfall Events Modify Trace Gas Concentrations in Central Amazonia"

**Overview:**

This manuscript analyzes vertical mixing ratio profiles of trace gases measured in the Amazon Forest at the ATTO field site. To examine the effects that rainfall has on vertical profiles of trace gases, the measurements are grouped into times relative to the peak rainfall rate. The analysis then looks at the differences in concentrations at various heights before and after the peak rain fall, with the intent of highlighting processes such as vertical mixing, chemical reactions, emissions, and uptake (i.e. sources and sinks).

This is a valuable data set and analysis, and the results are interesting, but there are some concerns with the approach and the interpretation of the results which are outlined below.

**General Comments:**

1) Figure clarity must be improved. The font size in all the figures it too small to read. It is difficulty to distinguish individual lines in Figure 3. The colour scales should all be symmetrical around zero (so that zero is the same colour in each plot). And the panels in the supplementary figure appear to be out of order and unlabelled.

2) Concentration is used through to describe mixing ratios, except for one instance (343) where the measurements are referred to as "mole fraction".

3) Statistically, showing confidence intervals (CI) with a median value gives no way of interpreting statistical significance. The CI is the confidence in the calculated mean, not the median. If you want to show the variation in data, show the median and percentiles. If you want to show statistical significance to compare different means (which I think is the case here), you need to show mean and CI. If there is a problem with outliers influencing the mean value (which is perhaps why you are using medians), consider using a truncated mean instead.

4) The methodology of the composite analysis is poorly explained. Consider demonstrating this with some variables. If you define $t_0 = 0$ as the time of peak rainfall, then the median profile at $t_0$ (+/- ½ hour) is $C_0(z, t_0)$, and the differences (not fluctuations) are calculated for each height and time as $\Delta C(z, t) = C(z, t) - C_0(z, t_0)$, where $C(z, t)$ is the median vertical mixing ratio profile using all values at time $t$ (+/- ½ hour). If this is a correct interpretation of what was done, then $\Delta C(z, t_0)$ will be zero for all $z$ values. But this isn't the case in the figures (e.g. Fig. 4a, 5c,d,f, and 6a-d). More explanation of the process and why $\Delta C(z, t_0)$ isn't zero is needed.

5) While the requirement to reference all the data to a single common time (peak rainfall rate) is understandably necessary, it isn't clearly argued that this is the best choice. For example, if modellers want to use these results to improve predicted concentration measurements during and after rainfall events, how would this reference be useful for them? Why not use a certain minimum

threshold value so that the reference time is the start of the rainfall event?  The authors should at least provide some demonstration of analysis to show why peak rainfall rate is the better reference point.

6) No statistical analysis of the rain events is given.  The reader has no knowledge of how long the rains typically are or when they occur.  How does the 4-hour window shown in the figures compare to length of time of a typical rainfall?  Presumably, rainfall rate with time would be one of the most important variables for this analysis, but it is missing in discussion and presentation of meteorological variables.

7) There is a flaw in the interpretation of the results (at multiple points in the discussion) where higher values of $\Delta C$ when $t < 0$ are referred to as "increases" in concentration.  This seems to imply a causal relationship where rainfall near $t = 0$ results in an increase in concentration at $t < 0$.  While there might be some changes in the forest in "anticipation" of a rain event, this still seems like a misinterpretation.  In reality, the concentrations are decreasing from an initial pre-rain profile at $t < 2$ hours.  This again demonstrates why it would be better to set the reference point as the start of the rain event (or at the very least to use results to demonstrate why that isn't the better approach).

8) In all the cases shown, the rain results in some observable value of $\Delta C$ over a 4-hour period (although the statistical significance of the change in never demonstrated), which seems like it would continue past $t > 2$ hours.  How long a period would you need to analyze to demonstrate that the profile returns to its original shape (i.e. $\Delta C = 0$ for all $z$)?  If the rainfall events are evenly distributed in time, then it should be possible to show this.  But if rainfall events are only occurring during certain parts of the day (during an increase in temperature for example), then your analysis isn't separating out both effects.  If $\Delta C$ continues to change for many hours passed the rain event, then can you attribute the changes to the rain event at all?  Showing that $\Delta C = 0$ over some given time would give more confidence in the analysis.

**Specific Comments:**

1) The discussion of previous works looking at the effect of rain in the forest could be strengthened. The Gerkin et al. measurements of ozone are mentioned, but very little information is given. Additionally, Wang et al. and Sigler et al. are referenced later in the discussion, but there is no mention of these related works in the Intro.

2) Many subjective or overly descriptive terms are used in the manuscript.  These include "well-equipped" (27), "invaluable" (31), "extensively" (43), "precisely captured" (100), "state of the air instruments" (105), "consistently delivers data with remarkable precision" (106), "exceptional stability" (108), and "rigorous testing" (109).

3) There are also many instances where the language is too strong for the results shown. For example, I would request changing "point at" 279 to "suggest"; "directly linked" (282) to "correlated",

"probably due" (329) to "possibly due"; "indicates" (355) to "suggests"; "can" (379) to "could"; and "indicate" (412) to "suggest".

4) I could be wrong, but I don't think canopy height was given in the manuscript (although it can be roughly inferred from the figures). Please add an indication of canopy height to Figure 1 either as a dashed line or as an axis scale (on the right) of z/hc.

5) At (114) how many is "several"?

6) For the met variables (121-136), give all the measurement heights.

7) At (144) what is meant by "This study"?

8) At (194) what is meant by "leading"?

9) At (224) what is meant by "particular"?

10) In the discussion of Rn activity (236 to 240), the confidence intervals (CI) shown in Fig. 2 suggest that the changes in time described in the text aren't statistically significant. To demonstrate that the changes are significant you would need to show dC/dt > the CI of dC/dt.

11) Lines (263 to 278) should go in the Introduction.

12) Add a reference for the sentence at (283-284).

13) At (288), do you mean "less important at night"?

14) At (296), it's not clear how you know "Under daylight conditions and before the rain event, CO profiles exhibit a strong source near the ground and show lower concentrations near the canopy". Fig. 4 shows $\Delta C$ not $C$. This is also the case for (307) "During the day, the overall picture changes completely with rainfall, shifting from a strong vertical gradient towards a relatively well-mixed layer from the ground to 79 m." and (311) "inversion of the profile". Because Fig. 4 just shows change, there is no way to know that the profile has inverted.

15) Lines (297 to 306) discuss CO emissions in general, but I don't see the connection to rain events.

16) At line (317) "throughout both day and night periods", refer back to Figure 1.

17) At line (337), where is standard deviation shown and what data set is it from?

18) At (372), how would NO2 be "washed out"?

19) At (384), I would suggest changing "As soon as rain starts…" to "Typically, when rain starts in a forest…" (to emphasize you are not discussing your data).

20) The discussion at Lines (390 to 396) is very speculative and should be rewritten without the speculation and subjective language (or removed completely).

**Technical Corrections:**

(Line 4): NO2 written twice.

(6): Remove "its"

(38): "measurements" should be "concentrations".

(47): "originating" would be better than "coming from".

(49) delete "of".

(67) Missing space before reference.

(75) What does "depend on and independent" mean?

(78) Put citation in brackets.

(87) "vertical profile measurements" might be better than "measurement profiles"

(88) "vertical profile"

(99) delete "respectively"

(105) delete "specifically with"

(109) Is the serial number relevant?

(113 + other locations) "instant" should be capitalised.

(169) "tipping amount" or "tipping threshold"

(224) replace "its" with "the nighttime"

Figure 3: New sentence at "Air temperature was measured at a height of…"

(349) "Figure 5…"

(352) "from" instead of "in"

(362) replace "as to be" with "is"

(411) Space before NO2.

(423) "variability patterns"

---

## Referee Comment (RC2)

Review of "How Rainfall Events Modify Trace Gas Concentrations in Central Amazonia" by Machado et al.

General

Machado et al. examined how rainfall events modify eight kinds of trace gas concentrations in an Amazon forest. The eight trace gases include $CO_2$, $CO$, $CH_4$, $O_3$, $NO$, $NO_2$, isoprene, and monoterpene. Their analysis is based on multiple year's measurement of these gases from the surface to 80 meters at tower in an Amazon forest. They divided these rainfall events into daytime and nighttime groups. They made composites for these two groups for each of the gases within a 4-hour window: from two hours before the peaking rainfall time to two hours after the peaking rainfall time. This involves a big effort in measurement and data analysis. The discussion is comprehensive. The presented results are valuable and can enhance our understanding of this research topic. Nevertheless, I have the following comments for the authors to consider when revising their paper.

The authors use a term, "fluctuation" of the trace gas concentrations, to show the rainfall impact without explaining how the fluctuation is defined. Is it the difference in the corresponding gas concentration at the time from the background concentration? If so, is the seasonal variation in the background concentration considered?

The authors provided the profiles in the daytime and nighttime for each gas during the rainfall event (Fig. 1). Can the authors also provide the background profiles without rainfall events?

I also believe that showing the actual ozone profiles during the rainfall events, in additional to their anomaly from the background, will help the authors to illustrate their points. Such profiles can be shown in the Supplement.

The authors used a 4-hour window that centres at "maximum rain rate". It is not clear how rainfall is distributed during the 4 hours. No rain at all except at the time with the maximum rain rate time?

As Figure 3 shows, the variation in rainfall is associated with changed in other meteorological elements (radiation, cloud cover, temperature, humidity, wind, boundary layer height, and GLM density). The authors discussed the impact of rainfall on the trace gas concentrations mainly based on the variations in other meteorological elements. The authors missed the discussion on direct rainfall impact on these gas concentrations through examining the solubility of these gases.

Line 282-290, "The decrease of CO2 concentration within the canopy after the rainfall is directly linked to the simultaneous increase in humidity and cloud cover and decrease in temperature". The reduction in radiation is likely to be the main driver for the variation in $CO_2$ concentration, this is not explicitly mentioned. "Another possible reason could be associated with an increase in mixing within the canopy, destroying the stable layer within the canopy by mixing free

tropospheric air into the canopy." Can the authors provide supporting evidence for this? This also applies for other discussions in the paper, the audience would be benefited if some pieces of supporting evidence are provided. If no supporting evidence, the authors can use phrases like: "we suspect", "this study suggests", or some expressions like that.

Minor

Line 245, 292, 327: Figure ??

Line 227, "Environmental", "E" should be in a lower case.

Line 259, "Carbon Monoxide", "C" and "M" should be in lower cases.

Line 349, Add "Fig." before "5".

Line 385, change "reported in (Pfannerstill et al. 2021)" to "reported in Pfannerstill et al. (2021)". Similarly, in Line 392 and other places.

Line 348, can this reference be cited in this way? "Machado, L. and all: How the Amazonian Forest Produces New Particles, Submitted to Nature, XX, XX, 2023."

Fonts for some figures are too small to read.

---

## Author Comment (AC1)

**REFERENCE - ACP submission  DOI: -10.5194/egusphere-2023-2901 – " How Rainfall Events Modify Trace Gas Concentrations in Central Amazonia" by Machado et al.**

Dear Editor Dra. Graciela Raga,

Thank you for considering our manuscript "How Rainfall Events Modify Trace Gas Concentrations in Central Amazonia" for publication in ACP. Please find attached the revised manuscript. We are grateful to both reviewers for their constructive, detailed, insightful and helpful reviews, which helped us to improve our manuscript. Below, we provide a point-by-point response to the comments, concerns and suggestions made by both reviewers, and also outline the changes made in the revised manuscript. We hope that you will find our revisions satisfactory. The reviewers' comments are in black and our responses are in blue in regular font; changes to the manuscript text are in *blue italics and underlined*.

**Referee #1 (Remarks to the Author):**

**Manuscript format description:**

Black text shows the original referee comment, blue text shows the authors response. Changes to the manuscript text are shown as *italicized and underlined*. We used bracketed comment numbers for referee comments (e.g., [R1.1]) and author's responses (e.g., [A1.1]).

Dear reviewer, your detailed review was very important for the improvement of the article. Thank you for taking the time to review our manuscript and provide constructive feedback to improve our manuscript.

**General Comments:**

[R1.1]. Figure clarity must be improved. The font size in all the figures it too small to read. It is difficulty to distinguish individual lines in Figure 3. The colour scales should all be symmetrical around zero (so that zero is the same colour in each plot). And the panels in the supplementary figure appear to be out of order and unlabelled.

[A1.1]. Thanks for the recommendations, Figure 3 font size and line width were increased. Panels in the Supplementary Figures were corrected labelled and reordered. Composite are presented with the similar color bar, with zero always with the same color.

[R1.2].  Concentration is used through to describe mixing ratios, except for one instance (343) where the measurements are referred to as "mole fraction".

[A1.2]. Thank you for correcting this, it was changed to the correct denomination in the manuscript.

[R1.3]. Statistically, showing confidence intervals (CI) with a median value gives no way of interpreting statistical significance. The CI is the confidence in the calculated mean, not the median. If you want to show the variation in data, show the median and percentiles. If you want to show statistical significance to compare different means (which I think is the case here), you need to show mean and CI. If there is a problem with outliers influencing the mean value (which is perhaps why you are using medians), consider using a truncated mean instead.

[A1.3]. Thanks, we now present the median and the percentiles of 30% and 70%.

[R1.4]. The methodology of the composite analysis is poorly explained. Consider demonstrating this with some variables. If you define $tt0 = 0$ as the time of peak rainfall, then the median profile at $tt0$ (+/- ½ hour) is $CC0(zz, tt0)$, and the differences (not fluctuations) are calculated for each height and time as $\Delta CC(zz, tt) = CC(zz, tt) - CC0(zz, tt0)$, where $CC(zz, tt)$ is the median vertical mixing ratio profile using all values at time $tt$ (+/- ½ hour). If this is a correct interpretation of what was done, then $\Delta CC(zz, tt0)$ will be zero for all $zz$ values. But this isn't the case in the figures (e.g. Fig. 4a, 5c,d,f, and 6a-d). More explanation of the process and why $\Delta CC(zz, tt0)$ isn't zero is needed.

[A1.4]. Thanks for the suggestions, we added an explanation on how the calculation were done. There is no zero for all zz values at tt0 because the median value was not done for tt0. Median Mixing ration is time-independent, and it was calculated for all times when the rain rate was larger than 0,5 mm/hr. That is why it is important to explain the calculations done, thanks again. We added this information to the manuscript:

"*To compute the composites, we first define the median mixing ratio during the rain event as the median value of the trace gas C at the height z during all times when rain was observed during the day (Cday(z)median), and the night (Cnight(z)median). The composite was constructed as the median at each time (t) corresponding to the window between 15 minutes before and 15 minutes after time t. The composite was performed for each trace gas, for day and night, for the time between 2 hours before and 2 hours after the maximum rainfall, every 30 minutes. The Mixing ratio composite difference is then calculated as: ΔCday=night(z, t) = Cday=night(z,t) - Cday=night(z)median). The composite was defined in this way to highlight the variability of the gas mixing ratio during the rainfall event.*"

[R1.5]. While the requirement to reference all the data to a single common time (peak rainfall rate) is understandably necessary, it isn't clearly argued that this is the best choice. For example, if modellers want to use these results to improve predicted concentration measurements during and after rainfall events, how would this reference be useful for them? Why not use a certain minimum threshold value so that the reference time is the start of the rainfall event? The authors should at least provide some demonstration of analysis to show why peak rainfall rate is the better reference point.

[A1.5]. We tried different ways of doing the composite. We used the beginning, the end and the maximum rainfall. Each method has a specific limitation, but they all have the same problem, the rain events have different durations, therefore it is not possible to establish a

normalization of the rain event duration using any of these different possibilities. If the beginning of the rain event is chosen, the end is not fixed, if the end of the event is chosen, the beginning is not fixed. Therefore, we decided to choose the time of the maximum rain rate, to focus on the rain effect, referring to the time when the rain reached the maximum intensity, so that the effect we are looking for could be referred to as the time t0. The modellers should consider the different effects of rain on gas species, compare to the simulation results, and apply new parameterizations or dynamic processes. Our results will not provide information on how the atmosphere returns to the background state. We have added examples of the different types of composites and a discussion of the choice of maximum rainfall rate. In fact, the results are very similar as about 64% of the cases have durations of less than 120 minutes.

[R1.6]. No statistical analysis of the rain events is given. The reader has no knowledge of how long the rains typically are or when they occur. How does the 4-hour window shown in the figures compare to length of time of a typical rainfall? Presumably, rainfall rate with time would be one of the most important variables for this analysis, but it is missing in discussion and presentation of meteorological variables.

[A1.6].We introduced the rain events statistics in order to provide a background discussion about the rain events duration, and rain rate. The new Figure was added as a complement to the Figure 3. A discussion about rain rate and rainfall duration was added along the text.

[R1.7]. There is a flaw in the interpretation of the results (at multiple points in the discussion) where higher values of $\Delta CC$ when $tt < 0$ are referred to as "increases" in concentration. This seems to imply a causal relationship where rainfall near $tt =0$ results in an increase in concentration at $tt < 0$. While there might be some changes in the forest in "anticipation" of a rain event, this still seems like a misinterpretation. In reality, the concentrations are decreasing from an initial pre-rain profile at $tt < 2$ hours. This again demonstrates why it would be better to set the reference point as the start of the rain event (or at the very least to use results to demonstrate why that isn't the better approach).

[A1.7]. We review the discussion and correct any interpretation that considers any absolute value of mixing ratio, we added a discussion about the composite time and it is clear now for the reader how composite was done, the rainfall duration, the nighttime and daytime variability and the mixing ratio profiles for rain and no-rain events. Figures were added to the supplement.

[R1.8]. In all the cases shown, the rain results in some observable value of $\Delta CC$ over a 4-hour period (although the statistical significance of the change in never demonstrated), which seems like it would continue past $tt > 2$ hours. How long a period would you need to analyze to demonstrate that the profile returns to its original shape (i.e. $\Delta CC = 0$ for all $zz$)? If the rainfall events are evenly distributed in time, then it should be possible to show this. But if rainfall events are only occurring during certain parts of the day (during an increase in temperature for example), then your analysis isn't separating out both effects. If $\Delta CC$ continues to change for many hours passed the rain event, then can you attribute the changes to the rain event at all? Showing that $\Delta CC = 0$ over some given time would give more confidence in the analysis.

[A1.8]. We cannot control the time of rain, the rain rate and the rainfall duration. The results should be interpreted as the changes before, at the time of maximum rain rate and after, as a

composition of different rain types. However, most of the rain events start and end inside the same hour, therefore the results are biased by this short rainfall duration. Further studies could compute the evolution for different rain rates, rain duration, season, hour of the day. This study has the main goal to show how each of the 8 analyzed species change as function of the rainfall in a more holistic view. To give an idea of the significance of the differences in gas variation during the rain event, we have added a supplementary figure showing the diurnal cycle for each gas for the rainy and non-rainy events. The figure gives a clear picture of the difference in gas mixing ratios between the two cases.

***Specific minor comments:***

[R1.9] The discussion of previous works looking at the effect of rain in the forest could be strengthened. The Gerkin et al. measurements of ozone are mentioned, but very little information is given. Additionally, Wang et al. and Sigler et al. are referenced later in the discussion, but there is no mention of these related works in the Intro.

[A1.9]. A discussion including these papers was added to the Introduction as requested.
"*Ozone concentration is strongly modulated by precipitation, as observed by Betts et al. (2002) during the Large-Scale Biosphere Atmosphere Experiment in Amazonia wet season experiment. They found an increase in ozone concentration and a decrease in potential temperature as an indication of convective downdraft. They suggested the important role of the ozone increase in the photochemical process in the boundary layer. Sigler et al. (2002) studied the effect of ozone on forested and deforested regions in Amazonas. They found higher ozone concentrations over pasture than over forest, suggesting that the forest has a more effective sink mechanism to consume ozone. Gerken et al. (2016) studied ozone dynamics during the GoAmazon experiment. They compared concentrations between the dry and wet seasons and found that the average concentration differed by 5 ppbv, higher during the dry season with an average concentration of 20 ppbv. They observed peaks of up to 25 ppbv in the boundary layer during rain events. Lighting can also increase $O_3$ mixing ratios, as discussed by Shlanta and Moore (1972).*"

[R1.10] Many subjective or overly descriptive terms are used in the manuscript. These include "well equipped" (27), "invaluable" (31), "extensively" (43), "precisely captured" (100), "state of the air instruments" (105), "consistently delivers data with remarkable precision" (106), "exceptional stability" (108), and "rigorous testing" (109).

[A1.10]. These terms have been deleted or modified in the manuscript.

[R1.11] There are also many instances where the language is too strong for the results shown. For example, I would request changing "point at" 279 to "suggest"; "directly linked" (282) to "correlated","probably due" (329) to "possibly due"; "indicates" (355) to "suggests"; "can" (379) to "could"; and "indicate" (412) to "suggest".

[A1.11]. Thanks for the suggestions, it was changed as recommended.

[R1.12] I could be wrong, but I don't think canopy height was given in the manuscript (although it can be roughly inferred from the figures). Please add an indication of canopy height to Figure 1 either as a dashed line or as an axis scale (on the right) of z/hc.

[A1.12]. The height of the canopy has been included in the text and a line at the height of the canopy has been included in Figure 1.

[R1.13] At (114) how many is "several"?

[A1.13].We changed the sentence.

[R1.14] For the met variables (121-136), give all the measurement heights.

[A1.14]. It was included in the text.

[R1.15]  At (144) what is meant by "This study"?

[A1.15]. It was clarified in the text; it is related to the measurements of the RN 222.

[R1.16] At (194) what is meant by "leading"?

[A1.16]. We corrected the sentence. "*The gas profiles shown in Figure 1 serve as the baseline for the composite analysis, highlighting variations in the profiles around (4-hour window) the precipitation events. Composites were derived by calculating deviations from the median profile within a 4-hour window, covering 2 hours before and 2 hours after the peak rain rate.*"

[R1.17] At (224) what is meant by "particular"?

[A1.17]. The sentence was modified – "*In contrast, monoterpenes accumulate in the canopy at night and could be released during rain events. Although their mixing ratio is lower than during the day, their vertical distribution remains constant. The monoterpene production could be influenced by mechanical turbulence within the vegetation, especially during rainfall events, and modulated by the air temperature..*"

[R1.18] In the discussion of Rn activity (236 to 240), the confidence intervals (CI) shown in Fig. 2 suggest that the changes in time described in the text aren't statistically significant. To demonstrate that the changes are significant you would need to show $dC/dt$ > the CI of $dC/dt$.

[A1.18]. We added a comment in these paragraphs explain that the variation is not statistically significant. "*We should observe a large variability during the night due to the small sample; the large variability of the sampling and the wide confidence interval do not allow us to say that these changes in behavior associated with rainfall are statistically significant. However, we tested the behavior separately for only 2019 and 2020, and the pattern of maximum activity concentration before maximum rainfall is consistent between the two years.*"

[R1.19] Lines (263 to 278) should go in the Introduction.

[A1.19]. It was changed as recommended.

[R1.20] Add a reference for the sentence at (283-284).

[A1.20]. A reference was added *(Pedruzo-Bagazgoitia, X., Patton, E. G., Moene, A. F., Ouwersloot, H. G., Gerken, T., Machado, L. A. T., Martin, S. T., Sörgel, M., Stoy, P. C., Yamasoe, M. A., and Vilà-Guerau de Arellano, J.: Investigating the Diurnal Radiative, Turbulent, and Biophysical Processes in the Amazonian Canopy-Atmosphere Interface by Combining LES Simulations and Observations, Journal of Advances in Modeling Earth Systems, 15, e2022MS003 210, 2023.)*

[R1.21] At (288), do you mean "less important at night"?

[A1.21]. We added at night to make the text clearer.

[R1.22] At (296), it's not clear how you know "Under daylight conditions and before the rain event, CO profiles exhibit a strong source near the ground and show lower concentrations near the canopy". Fig. 4 shows $\Delta CC$ not $CC$. This is also the case for (307) "During the day, the overall picture changes completely with rainfall, shifting from a strong vertical gradient towards a relatively well-mixed layer from the ground to 79 m." and (311) "inversion of the profile". Because Fig. 4 just shows change, there is no way to know that the profile has inverted.

[A1.22]. The parts of the text that have interpretation related to the absolute value was changed as recommended.

[R1.23] Lines (297 to 306) discuss CO emissions in general, but I don't see the connection to rain events.

[A1.23].We understand that a large discussion about the sources and sinks of the CO is useful as the processes are more complexes than the others gases (as well as the CH4).

[R1.24] At line (317) "throughout both day and night periods", refer back to Figure 1.

[A1.24]. Changed as recommended.

[R1.25] At line (337), where is standard deviation shown and what data set is it from?

[A1.25]. Thanks for the comment, the sentence was changed.

[R1.26] At (372), how would NO2 be "washed out"?

[A1.26]. There are manuscript describing the washed out effect. Martins (1984, Atmospheric Environment Vol. 18. No. 9. pp. 1955-1961. 1984. ESTIMATED WASHOUT COEFFICIENTS FOR SULPHUR DIOXIDE, NITRIC OXIDE, NITROGEN DIOXIDE AND OZONE) that describes the washout effect for NO2. Reference was added to the text.

[R1.27] At (384), I would suggest changing "As soon as rain starts…" to "Typically, when rain starts in a forest…" (to emphasize you are not discussing your data). The discussion at Lines (390 to 396) is very speculative and should be rewritten without the speculation and subjective language (or removed completely).

[A1.27]. We changed as recommended. However, we considered that the discussion on page 390-396 is important to present one explanation for the Monoterpene behavior at night. It is not speculative, as it is based on clear research results cited in the text.

[R1.28] (49) (Line 4): NO2 written twice.

[A1.28]. Corrected, thanks.

[R1.29] (6): Remove "its"

[A1.29]. Changed as recommended

[R1.30] (38): "measurements" should be "concentrations".

[A1.30].Changed as recommended

[R1.31] (47): "originating" would be better than "coming from".

[A1.31]. Changed as recommended

[R1.32] (49) delete "of".

[A1.32]. Changed as recommended

[R1.33] (67) Missing space before reference.

[A1.33]. Changed as recommended

[R1.34] (75) What does "depend on and independent" mean?

[A1.34]. The text was changed – "*Their release to the atmosphere depends on solubility and volatility and may, therefore, be a function of physiological gas exchange regulation under stomatal control. Some of the BVOC species are released close to the mixing ratio gradient between outside air and plant tissue; some are under strict stomatal control. This behavior strongly depends on water solubility, i.e., equilibrium gas–aqueous phase partition coefficient (Niinemets, 2007)..*"

[R1.35] (78) Put citation in brackets.

[A1.35]. Changed As recommended

[R1.36] (87) "vertical profile measurements" might be better than "measurement profiles"

[A1.36]. Changed as recommended

[R1.37] (88) "vertical profile"

[A1.37]. Changed as recommended

[R1.38] (99) delete "respectively"

[A1.38]. Changed as recommended

[R1.39] (105) delete "specifically with"

[A1.39].Changed as recommended

[R1.40] (109) Is the serial number relevant?

[A1.40]. No, it was deleted.

[R1.41] (113 + other locations) "instant" should be capitalised.

[A1.41]. Changed as recommended

[R1.42] (169) "tipping amount" or "tipping threshold"

[A1.42].Thanks we added tipping amount

[R1.43] (224) replace "its" with "the nighttime"

[A1.43]. The sentence was modified based in comment R1.17.

[R1.44] Figure 3: New sentence at "Air temperature was measured at a height of…"

[A1.44]. It was changed as recommended.

[R1.45] (349) "Figure 5…"

[A1.45]. It was corrected.

[R1.46] (352) "from" instead of "in"

[A1.46]. Changed as recommended

[R1.47] (362) replace "as to be" with "is"

[A1.47].Changed as recommended

[R1.48] (411) Space before NO2.

[A1.48]. Changed as recommended

[R1.49] (423) "variability patterns

[A1.49]. Changed as recommended

---

## Author Comment (AC2)

**REFERENCE - ACP submission  DOI: -10.5194/egusphere-2023-2901 – " How Rainfall Events Modify Trace Gas Concentrations in Central Amazonia" by Machado et al.**

Dear Editor Dra. Graciela Raga,

Thank you for considering our manuscript "How Rainfall Events Modify Trace Gas Concentrations in Central Amazonia" for publication in ACP. Please find attached the revised manuscript. We are grateful to both reviewers for their constructive, detailed, insightful and helpful reviews, which helped us to improve our manuscript. Below, we provide a point-by-point response to the comments, concerns and suggestions made by both reviewers, and also outline the changes made in the revised manuscript. We hope that you will find our revisions satisfactory. The reviewers' comments are in black and our responses are in blue in regular font; changes to the manuscript text are in *blue italics and underlined*.

**Referee #2 (Remarks to the Author):Review for manuscript entitled, " How Rainfall Events Modify Trace Gas Concentrations in Central Amazonia " by Machado et al**

**Manuscript format description:**

Black text shows the original referee comment, blue text shows the authors response. Changes to the manuscript text are shown as *italicized and underlined*. We used bracketed comment numbers for referee comments (e.g., [R2.1]) and author's responses (e.g., [A2.1]).

**General Comments:**

[R2.1] The authors use a term, "fluctuation" of the trace gas concentrations, to show the rainfall impact without explaining how the fluctuation is defined. Is it the difference in the corresponding gas concentration at the time from the background concentration? If so, is the seasonal variation in the background concentration considered?

[A2.1] We used fluctuation incorrectly, the correct denomination is difference. The presented composite is computed as the difference between the average median value for all time steps, before and after the max rain rate, and composite value. We are now using difference in all text, including in the figures legend. We also added an explanation on how the composite was computed.

[R2.2] The authors provided the profiles in the daytime and nighttime for each gas during the rainfall event (Fig. 1). Can the authors also provide the background profiles without rainfall events?

[A2.2] We added supplementary figures showing rain and non-rain events for day, night, wet and dry season, for each gas mixing ratio profile. We also added in the supplement the diurnal cycle for rainy and non-rainy events.

[R2.3] I also believe that showing the actual ozone profiles during the rainfall events, in additional to their anomaly from the background, will help the authors to illustrate their points. Such profiles can be shown in the Supplement.

[A2.3] We added several new figures on the supplement showing the profiles for rain and non-rain events, we hope these new figures provide the information requested by the reviewer. We also added a new figure, with the composite of the absolute mixing ratio of the ozone, for different composite types.

[R2.4] The authors used a 4-hour window that centres at "maximum rain rate". It is not clear how rainfall is distributed during the 4 hours. No rain at all except at the time with the maximum rain rate time?

[A2.4] A discussion including new figures was added to the manuscript including information about rain duration, and intensity. Most of the rain events have a duration smaller than one hour.

[R2.5] As Figure 3 shows, the variation in rainfall is associated with changed in other meteorological elements (radiation, cloud cover, temperature, humidity, wind, boundary layer height, and GLM density). The authors discussed the impact of rainfall on the trace gas concentrations mainly based on the variations in other meteorological elements. The authors missed the discussion on direct rainfall impact on these gas concentrations through examining the solubility of these gases.

[A2.5] A discussion about solubility was included in the same paragraph mentioned by the reviewer. "*Rainfall is correlated with changes in other meteorological variables, as described above, but gas solubility is also directly affected by rain events. Rain can increase the rate of air-water gas exchange, Ho et al. (1997) empirically calculated the gas transfer velocity and rainfall rate for different rainfall rates and drop sizes, quantifying the enhancement of air-water gas exchange by rainfall. There is also the wet deposition effect, which may not be an efficient removal mechanism for hydrophobic gases as described by Mullaugh et al. (2015). The direct rainfall effect depends on the solubility of gases, and wet deposition is highly complex, especially for VOCs, due to the water solubility of this heterogeneous mixture covering several orders of magnitude (Niinemets and Reichstein, 2003).*"

[R2.6] Line 282-290, "The decrease of CO2 concentration within the canopy after the rainfall is directly linked to the simultaneous increase in humidity and cloud cover and decrease in temperature". The reduction in radiation is likely to be the main driver for the variation in CO2 concentration, this is not explicitly mentioned. "Another possible reason could be associated with an increase in mixing within the canopy, destroying the stable layer within the canopy by mixing free tropospheric air into the canopy." Can the authors provide supporting evidence for this? This also applies for other discussions in the paper, the audience would be benefited if some pieces of supporting evidence are provided. If no supporting evidence, the authors can use phrases like: "we suspect", "this study suggests", or some expressions like that.

[A2.6] The sentence was modified to "*The decrease in the CO2 mixing ratio within the canopy during the rain event is correlated with the simultaneous increase in humidity and decrease in temperature as a consequence of the reduction in radiation due to the increase in cloud cover Pedruzo-Bagazgoitia et al. (2023). As discussed above, these environmental conditions suppress both soil and tree CO2 exchange and surface flux and reduce photosynthesis. Another possible reason could be associated with increased mixing within the canopy, destroying the*

*stable layer within the canopy by mixing free tropospheric air into the canopy Betts et al. (2002). These two effects may contribute to the reduction in CO2 mixing ratio after the rain event; however, the importance of each of these effects could not be quantified with the current data.*".

**Specific minor comments:**

[R2.7] Line 245, 292, 327: Figure ??

[A2.7] All supplementary Figures was cited as ?? because the journal asked to split main text and supplementary in different text, and we didn't realize that problem. Sorry, now it was fixed.

[R2.8] Line 227, "Environmental", "E" should be in a lower case.

[A2.8] It was changed as recommended.

[R2.9] Line 259, "Carbon Monoxide", "C" and "M" should be in lower cases.

[A2.9] It was changed as recommended.

[R2.10] Line 349, Add "Fig." before "5".

[A2.10] It was changed as recommended.

[R2.11] Line 385, change "reported in (Pfannerstill et al. 2021)" to "reported in Pfannerstill et al.(2021)". Similarly, in Line 392 and other places.

[A2.11] It was changed as recommended

[R2.12] Line 348, can this reference be cited in this way? "Machado, L. and all: How the Amazonian Forest Produces New Particles, Submitted to Nature, XX, XX, 2023."

[A2.12] The reference was deleted as it is still in the review phase.

[R2.13] Fonts for some figures are too small to read.

[A2.13] The Figures fonts were changed.